# Lookup Table meets Local Laplacian Filter: Pyramid Reconstruction Network for Tone Mapping

**Feng Zhang** [1]    **Ming Tian** [1]    **Zhiqiang Li** [2]    **Bin Xu** [2]
**Qingbo Lu** [2]    **Changxin Gao** [1,3*]    **Nong Sang** [1]

[1] National Key Laboratory of Multispectral Information Intelligent Processing Technology,
School of Artificial Intelligence and Automation, Huazhong University of Science and Technology,
[2] DJI Technology Co., Ltd, [3] Hubei Key Laboratory of Brain-inspired Intelligent Systems
{fengzhangaia, tianming, cgao, nsang}@hust.edu.cn,
{mila.xu, cristopher.li, qingbo.lu}@dji.com

## Abstract

Tone mapping aims to convert high dynamic range (HDR) images to low dynamic range (LDR) representations, a critical task in the camera imaging pipeline. In recent years, 3-Dimensional Look-Up Table (3D LUT) based methods have gained attention due to their ability to strike a favorable balance between enhancement performance and computational efficiency. However, these methods often fail to deliver satisfactory results in local areas since the look-up table is a global operator for tone mapping, which works based on pixel values and fails to incorporate crucial local information. To this end, this paper aims to address this issue by exploring a novel strategy that integrates global and local operators by utilizing closed-form Laplacian pyramid decomposition and reconstruction. Specifically, we employ image-adaptive 3D LUTs to manipulate the tone in the low-frequency image by leveraging the specific characteristics of the frequency information. Furthermore, we utilize local Laplacian filters to refine the edge details in the high-frequency components in an adaptive manner. Local Laplacian filters are widely used to preserve edge details in photographs, but their conventional usage involves manual tuning and fixed implementation within camera imaging pipelines or photo editing tools. We propose to learn parameter value maps progressively for local Laplacian filters from annotated data using a lightweight network. Our model achieves simultaneous global tone manipulation and local edge detail preservation in an end-to-end manner. Extensive experimental results on two benchmark datasets demonstrate that the proposed method performs favorably against state-of-the-art methods.

## 1   Introduction

Modern cameras, despite their advanced and sophisticated sensors, are limited in their ability to capture the same level of detail as the human eye in a given scene. In order to capture more detail, high dynamic range (HDR) imaging techniques [25, 2] have been developed to convey a wider range of contrasts and luminance values than conventional low dynamic range (LDR) imaging. However, most modern graphics display devices have a limited dynamic range that is inadequate to reproduce the full range of light intensities present in natural scenes. To tackle this issue, tone mapping techniques [28, 34, 9] have been proposed to render high-contrast scene radiance to the

---

*Changxin Gao is the corresponding author, email: cgao@hust.edu.cn

37th Conference on Neural Information Processing Systems (NeurIPS 2023).

displayable range while preserving the image details and color appearance important to appreciate the original scene content.

Traditional tone mapping operators can be classified according to their processing as global or local. Global operators [11, 34, 7, 33, 21] map each pixel according to its global characteristics, irrespective of its spatial localization. This approach entails calculating a single matching luminance value for the entire image. As a result, the processing time is considerably reduced, but the resulting image may exhibit fewer details. In contrast, local operators [6, 8, 10, 24, 29] consider the spatial localization of each pixel within the image and process them accordingly. In essence, this method calculates the luminance adaptation for each pixel based on its specific position. Consequently, the resulting image becomes more visually accessible to the human eye and exhibits enhanced details, albeit at the expense of longer processing times. However, these traditional operators often require manual tuning by experienced engineers, which can be cumbersome since evaluating results necessitates testing across various scenes. Although system contributions have aimed to simplify the implementation of high-performance executables [32, 15, 27], they still necessitate programming expertise, incur runtime costs that escalate with pipeline complexity, and are only applicable when the source code for the filters is available. Therefore, seeking an automatic strategy for HDR image tone mapping is of great interest.

In recent years, there have been notable advancements in learning-based automatic enhancement methods [37, 12, 5, 30, 16, 35, 19], thanks to the rapid development of deep learning techniques [22]. Many of these methods focus on learning a dense pixel-to-pixel mapping between input high dynamic range (HDR) and output low dynamic range (LDR) image pairs. Alternatively, they predict pixel-wise transformations to map the input HDR image. However, most previous studies involve a substantial computational burden that exhibits a linear growth pattern in tandem with the dimensions of the input image.

To simultaneously improve the quality and efficiency of learning-based methods, hybrid methods [17, 41, 38, 36, 39] have emerged that combine the utilization of image priors from traditional operators with the integration of multi-level features within deep learning-based frameworks, leading to state-of-the-art performance. Among these methods, Zeng *et al.* [38] proposes a novel image-adaptive 3-Dimensional Look-Up Table (3D LUT) based approach, which exhibits favorable characteristics such as superior image quality, efficient computational processing, and minimal memory utilization. However, as indicated by the authors, utilizing the global (spatially uniform) tone mapping operators, such as the 3D look-up tables, may produce less satisfactory results in local areas. Additionally, this method necessitates an initial downsampling step to reduce network computations. In the case of high-resolution (4K) images, this downsampling process entails a substantial reduction factor of up to 16 times (typically downsampled to $256 \times 256$ resolution). Consequently, this results in a significant loss of image details and subsequent degradation in enhancement performance.

To alleviate the above problems, this work focuses on integrating global and local operators to facilitate comprehensive tone mapping. Drawing inspiration from the reversible Laplacian pyramid decomposition [3] and the classical local tone mapping operators, the local Laplacian filter [29, 1], we propose an effective end-to-end framework for the HDR image tone mapping task performing global tone manipulation while preserving local edge details. Specifically, we build a lightweight transformer weight predictor on the bottom of the Laplacian pyramid to predict the pixel-level content-dependent weight maps. The input HDR image is trilinear interpolated using the basis 3D LUTs and then multiplied with weighted maps to generate a coarse LDR image. To preserve local edge details and reconstruct the image from the Laplacian pyramid faithfully, we propose an image-adaptive learnable local Laplacian filter (LLF) to refine the high-frequency components while minimizing the use of computationally expensive convolution in the high-resolution components for efficiency. Consequently, we progressively construct a compact network to learn the parameter value maps at each level of the Laplacian pyramid and apply them to the remapping function of the local Laplacian filter. Moreover, a fast local Laplacian filter [1] is employed to replace the conventional local Laplacian filter [29] for computational efficiency. Extensive experimental results on two benchmark datasets demonstrate that the proposed method performs favorably against state-of-the-art methods.

In conclusion, the highlights of this work can be summarized into three points:

(1) We introduce an effective end-to-end framework for HDR image tone mapping. The network performs both global tone manipulation and local edge details preservation within the same model.

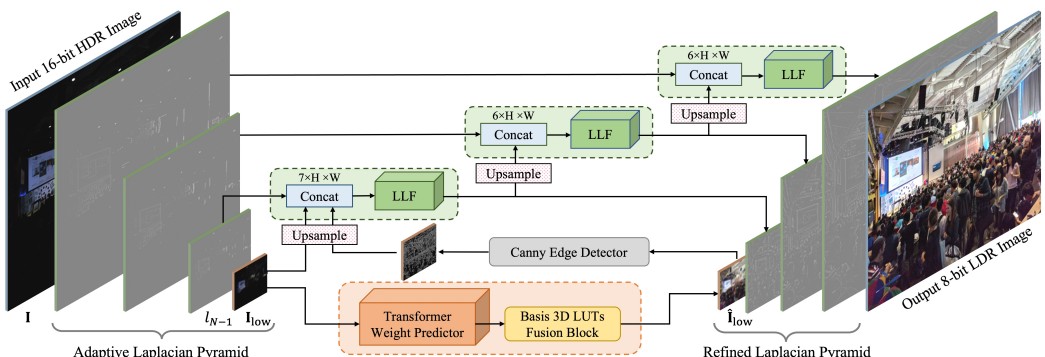

Figure 1: Overview of the framework. Our method first decomposes the input image $\mathbf{I}$ into a Laplacian pyramid. The low-frequency image $\mathbf{I}_{low}$ is fed into a lightweight transformer weight predictor and Basis 3D LUTs fusion block to transform into a low-resolution enhanced image $\hat{\mathbf{I}}_{low}$. To adaptively refine the high-frequency components, we progressively learn an image-adaptive local Laplacian filter (LLF) based on both high- and low-frequency images. Then, we perform the remapping function of the local Laplcian filter to refine the high-frequency components while preserving the pyramid reconstruction capability. For the level $N-1$, we concatenate the component with the edge map of $\hat{\mathbf{I}}_{low}$ to mitigate potential halo artifacts.

(2) We propose an image-adaptive learnable local Laplacian filter for efficient local edge details preservation, demonstrating remarkable effectiveness when integrated with image-adaptive 3D LUT.

(3) We conduct extensively experiments on two publically available benchmark datasets. Both qualitative and quantitative results demonstrate that the proposed method performs favorably against state-of-the-art methods.

## 2 Proposed Method

### 2.1 Framework Overview

We propose an end-to-end framework to manipulate tone while preserving local edge detail in HDR image tone mapping tasks. The pipeline of our proposed method is illustrated in Fig. 1. Given an input 16-bit HDR image $\mathbf{I} \in \mathbb{R}^{h \times w \times 3}$, we initially decompose it into an adaptive Laplacian pyramid, resulting in a collection of high-frequency components represented by $\mathbf{L} = [l_0, l_1, \cdots, l_{N-1}]$, as well as a low-frequency image denoted as $\mathbf{I}_{low}$. Here, $N$ represents the number of decomposition levels in the Laplacian pyramid. The adaptive Laplacian pyramid employs a dynamic adjustment of the decomposition levels to match the resolution of the input image. This adaptive process ensures that the low-frequency image $\mathbf{I}_{low}$ achieves a proximity of approximately $64 \times 64$ resolution. The described decomposition process possesses invertibility, allowing the original image to be reconstructed by incremental operations. According to Burt and Adelson [3], each pixel in the low-frequency image $\mathbf{I}_{low}$ is averaged over adjacent pixels by means of an octave Gaussian filter, which reflects the global characteristics of the input HDR image, including color and illumination attributes. Meanwhile, other high-frequency components contain edge-detailed textures of the image.

Motivated by the characteristics mentioned above of the Laplacian pyramid, we propose to manipulate tone on $\mathbf{I}_{low}$ while refining the high-frequency components $\mathbf{L}$ progressively to preserve local edge details. In addition, we progressively refine the higher-resolution component conditioned on the lower-resolution one. The proposed framework consists of three parts. Firstly, we introduce a lightweight transformer block to process $\mathbf{I}_{low}$ and generate content-dependent weight maps. These predicted weight maps are employed to fuse the basis of 3D LUTs. Subsequently, this adapted representation is used to transform $\mathbf{I}_{low}$ into $\hat{\mathbf{I}}_{low}$, resulting in the desired tone manipulation. Secondly, we construct parameter value maps by leveraging a learned model on the concatenation of $[l_{N-1}, up(\mathbf{I}_{low}), up(edge(\hat{\mathbf{I}}_{low}))]$, where $up(\cdots)$ represents a bilinear up-sampling operation and $edge(\cdots)$ denotes for the canny edge detector. These parameter value maps are then employed to perform a fast local Laplacian filter [1] on the Laplacian layer of level $N-1$. This step effectively

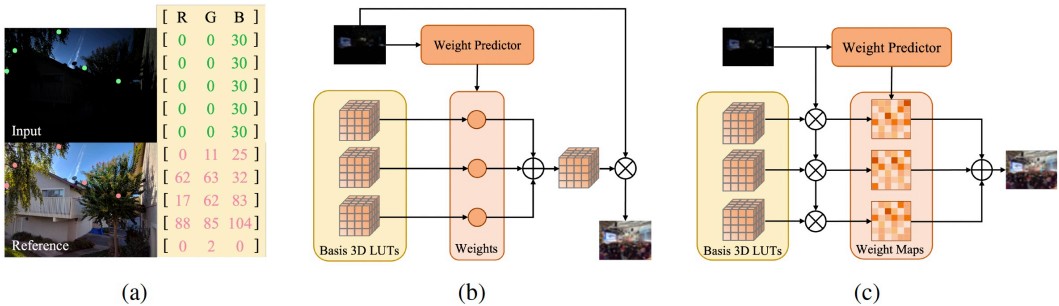

Figure 2: Illustration of the basis 3D LUTs fusion strategy. (a) present the multiple pixel mapping relationships of an image pair; (b) is the conventional basis 3D LUTs fusion strategy; (c) is the pixel-level basis 3D LUTs fusion strategy.

refines the high-frequency components while considering the local edge detail information. Lastly, we propose an efficient and progressive upsampling strategy to further enhance the refinement of the remaining Laplacian layers with higher resolutions. Starting from level $l = N - 2$ down to $l = 0$, we sequentially upsample the refined components from the previous level and concatenate them with the corresponding Laplacian layer. Subsequently, we employ a lightweight convolution block to perform another fast local Laplacian filter. This iterative process iterates across multiple levels, effectively refining the high-resolution components. We introduce these modules in detail in the following sections.

## 2.2  Pixel-level Basis 3D LUTs Fusion

According to the inherent properties of the Laplacian pyramid, the low-frequency image contains properties such as color and illumination of the images. Therefore, we employ 3D LUTs to perform tone manipulation on low-frequency images $\mathbf{I}_{low}$. In RGB color space, a 3D LUT defines a 3D lattice that consists of $N_b^3$ elements, where $N_b$ is the number of bins in each color channel. Each element defines a pixel-to-pixel mapping function $\mathbf{M}^c(i, j, k)$, where $i, j, k = 0, 1, \cdots, n_b - 1 \in \mathbb{I}_0^{N_b - 1}$ are elements' coordinates within 3D lattice and $c$ indicates color channel. Given an input RGB color $\{(\mathbf{I}_{(i,j,k)}^r, \mathbf{I}_{(i,j,k)}^g, \mathbf{I}_{(i,j,k)}^b)\}$, where $i, j, k$ are indexed by the corresponding RGB value, a output $\mathbf{O}^c$ is derived by the mapping function as follows:

$$\mathbf{O}_{(i,j,k)}^c = \mathbf{M}^c(\mathbf{I}_{(i,j,k)}^r, \mathbf{I}_{(i,j,k)}^g, \mathbf{I}_{(i,j,k)}^b). \tag{1}$$

The mapping capabilities of conventional 3D LUTs are inherently constrained to fixed transformations of pixel values. Fig. 2(a) demonstrates this limitation, where the input image has the same pixel values at different locations. However, these locations contain different pixel values in the reference image. While the input image is interpolated through a look-up table, the transformed image retains the same transformed pixel values at these locations. Consequently, the conventional 3D LUT framework fails to accommodate intricate pixel mapping relationships, thus impeding its efficacy in accurately representing such pixel transformations.

Inspired by [36], we propose an effective 3D LUT fusion strategy to address this inherent limitation. The conventional 3D LUT fusion strategy proposed by [38] is shown in Fig. 2(b), which first utilizes the predicted weights to fuse the multiple 3D LUTs into an image-adaptive one and then performs trilinear interpolation to transform images. In contrast, as shown in Fig. 2(c), our strategy is first to perform trilinear interpolation with each LUT and then fuse the enhanced image with predicted pixel-level weight maps. In this way, our method can enable a relatively more comprehensive and accurate representation of the complex pixel mapping relationships through the weight values of each pixel. The pixel-level mapping function $\mathbf{\Phi}^{h,w,c}$ can be described as follows:

$$\mathbf{O}_{(i,j,k)}^{h,w,c} = \mathbf{\Phi}^{h,w,c}(\mathbf{I}_{(i,j,k)}^r, \mathbf{I}_{(i,j,k)}^g, \mathbf{I}_{(i,j,k)}^b, \omega^{h,w}) = \sum_{n=0}^{N-1} \omega_n^{h,w} \mathbf{M}_n^c(\mathbf{I}_{(i,j,k)}^r, \mathbf{I}_{(i,j,k)}^g, \mathbf{I}_{(i,j,k)}^b), \tag{2}$$

where $\mathbf{O}_{(i,j,k)}^{h,w,c}$ is the final pixel-level output, $\omega_n^{h,w}$ represents a pixel-level weight map for $N$ 3D LUTs located at $(h, w)$. Note that our proposed strategy involves the utilization of multiple trilinear

interpolations, which may impact the computational speed when applied to high-resolution images. However, since our method operates at the resolution of $64 \times 64$, the computational overhead is insignificant. More discussions are provided in the supplementary material.

As shown in Fig. 1, given $\mathbf{I}_{low}$ with a reduced resolution, we feed it into a weight predictor to output the content-dependent weight maps $\omega^{h,w}$. Since the weight predictor aims to understand the global context, such as the brightness, color, and tone of an image, a transformer backbone is more suitable for extracting global information than a CNN backbone. Therefore, we utilize a tiny transformer model proposed by [23] as the weight predictor. The whole model contains only 400K parameters when $N = 3$.

## 2.3 Image-adaptive Learnable Local Laplacian Filter

Although the pixel-level basis 3D LUTs fusion strategy demonstrates stable and efficient enhancement of input images across various scenes, the transformation of pixel values through weight maps alone still falls short of significantly improving local detail and contrast. To tackle this limitation, one potential solution is to integrate a local enhancement method with 3D LUT. In this regard, drawing inspiration from the intrinsic characteristics of the Laplacian pyramid [3], which involves texture separation, visual attribute separation, and reversible reconstruction, the combination of 3D LUT and the local Laplacian filter [29] can offer substantial benefits.

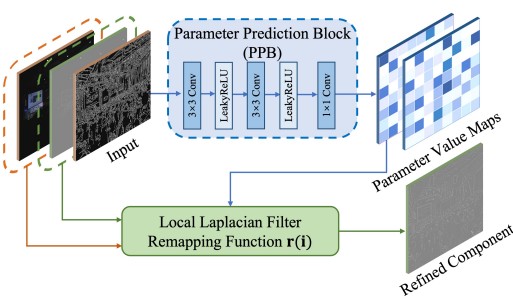

Figure 3: Architecture of the proposed image-adaptive learnable local Laplacian filter (LLF).

Local Laplacian filters are edge-aware local tone mapping operators that define the output image by constructing its Laplacian pyramid coefficient by coefficient. The computation of each coefficient $i$ is independent of the others. These coefficients are computed by the following remapping functions $\mathbf{r(i)}$:

$$\mathbf{r(i)} = \begin{cases} g + sign(i-g)\sigma_r(|i-g|/\sigma_r)^\alpha & if\ i \le \sigma_r \\ g + sign(i-g)(\beta(|i-g| - \sigma_r) + \sigma_r) & if\ i > \sigma_r \end{cases}, \tag{3}$$

where $g$ is the coefficient of the Gaussian pyramid at each level, which acts as a reference value, $sign(x) = x/|x|$ is a function that returns the sign of a real number, $\alpha$ is one parameter that controls the amount of detail increase or decrease, $\beta$ is another parameter that controls the dynamic range compression or expansion, and $\sigma_r$ defines the intensity threshold the separates details from edges.

Nevertheless, the conventional approach described in Eq. 3 necessitates manual parameter adjustment for each input image, leading to a cumbersome and labor-intensive process. To overcome this limitation, we propose an image-adaptive learnable local Laplacian filter (LLF) to learn the parameter value maps for the remapping function. The objective function of the learning scheme can be written as follows:

$$min_{\alpha,\beta}\mathcal{L}(\mathbf{r}(l,g), \mathbf{R}), \tag{4}$$

where $\alpha$ and $\beta$ are the learned parameter value maps of the Laplacian pyramid, $\mathcal{L}(\cdots)$ denotes the loss functions, $\mathbf{r}(l,g)$ presents the image-adaptive learnable local Laplacian filter (LLF), $l$ and $g$ are the coefficients of the Laplacian and Gaussian pyramid, respectively, $\mathbf{R}$ is the reference image. Note that the parameter $\sigma_r$ does not impact the filter's performance; thus, it is fixed at 0.1 in this paper. Furthermore, to enhance computational efficiency, we have employed the fast local Laplacian filter [1] instead of the conventional local Laplacian filter.

As discussed in Sec. 2.1, we have $l_{N-1} \in \mathbb{R}^{\frac{h}{2^{N-1}} \times \frac{w}{2^{N-1}} \times 3}$ and $\mathbf{I}_{low}, \hat{\mathbf{I}}_{low} \in \mathbb{R}^{\frac{h}{2^N} \times \frac{w}{2^N} \times 3}$. To address potential halo artifacts, we initially employ a Canny edge detector with default parameters to extract the edge map of $\hat{\mathbf{I}}_{low}$. Subsequently, we upsample $\mathbf{I}_{low}$ and $edge(\hat{\mathbf{I}}_{low})$ using bilinear operations to match the resolution of $l_{N-1}$ and concatenate them. The concatenated components are fed into a Parameter Prediction Block (PPB) as depicted in Fig. 3. The outputs of the PPB are utilized for the

| Methods | #Params | HDR+ (480p) | | | | HDR+ (original) | | | |
|---|---|---|---|---|---|---|---|---|---|
| | | PSNR↑ | SSIM↑ | LPIPS↓ | △E↓ | PSNR↑ | SSIM↑ | LPIPS↓ | △E↓ |
| UPE [35] | 999K | 23.33 | 0.852 | 0.150 | 7.68 | 21.54 | 0.723 | 0.361 | 9.88 |
| HDRNet [12] | 482K | 24.15 | 0.845 | 0.110 | 7.15 | 23.94 | 0.796 | 0.266 | 6.77 |
| CSRNet [14] | **37K** | 23.72 | 0.864 | 0.104 | 6.67 | 22.54 | 0.766 | 0.284 | 7.55 |
| DeepLPF [26] | 1.72M | 25.73 | 0.902 | 0.073 | 6.05 | N.A. | N.A. | N.A. | N.A. |
| LUT [38] | 592K | 23.29 | 0.855 | 0.117 | 7.16 | 21.78 | 0.772 | 0.303 | 9.45 |
| sLUT [36] | 4.52M | 26.13 | 0.901 | 0.069 | 5.34 | 23.98 | 0.789 | 0.242 | 6.85 |
| CLUT [39] | 952K | 26.05 | 0.892 | 0.088 | 5.57 | 24.04 | 0.789 | 0.245 | 6.78 |
| **Ours** | 731K | **26.62** | **0.907** | **0.063** | **5.31** | **25.32** | **0.849** | **0.149** | **6.03** |

Table 1: Quantitative comparison on HDR+ [13] dataset. "N.A." means that the results are not available due to insufficient memory of the GPU.

remapping function $\mathbf{r}(\mathbf{i})$ to refine $l_{N-1}$:

$$\hat{l}_{N-1} = \mathbf{r}(l_{N-1}, g_{N-1}, \alpha_{N-1}, \beta_{N-1}). \tag{5}$$

Subsequently, we adopt a progressive upsampling strategy to match the refined high-frequency component $\hat{l}_{N-1}$ with the remaining high-frequency components. This upsampled component is concatenated with $l_{N-2}$. As depicted in Fig. 1, the concatenated vector $[l_{N-2}, up(\hat{l}_{N-1})]$ is feed into another LLF. The refinement process continues iteratively, progressively upsampling until $\hat{l}_0$ is obtained. By applying the same operations as described in Eq. 5, all high-frequency components are effectively refined, leading to a set of refined components $[\hat{l}_0, \hat{l}_1, \ldots, \hat{l}_{N-1}]$. Finally, the result image $\hat{\mathbf{I}}$ is reconstructed using the tone mapped image $\hat{\mathbf{I}}_{low}$ with refined components $[\hat{l}_0, \hat{l}_1, \ldots, \hat{l}_{N-1}]$.

## 2.4 Overall Training Objective

The proposed framework is trained in a supervised scenario by optimizing a reconstruction loss. To encourage a faithful global and local enhancement, given a set of image pairs $(\mathbf{I}, \mathbf{R})$, where $\mathbf{I}[i]$ and $\mathbf{R}[i]$ denote a pair of 16-bit input HDR and 8-bit reference LDR image, we define the reconstruction loss function as follows:

$$\mathcal{L}_1 = \sum_{i=1}^{H \times W} (\| \hat{\mathbf{I}}[i] - \mathbf{R}[i] \|_1 + \| \hat{\mathbf{I}}_{low}[i] - \mathbf{R}_{low}[i] \|_1), \tag{6}$$

where $\hat{\mathbf{I}}[i]$ is the output of network with $\mathbf{I}[i]$ as input, $\hat{\mathbf{I}}_{low}[i]$ is the output of 3D LUT with $\mathbf{I}_{low}[i]$ as input, $\mathbf{R}_{low}[i]$ is the low-frequency image of reference image $\mathbf{R}[i]$.

To make the learned 3D LUTs more stable and robust, some regularization terms from [38], including smoothness term $\mathcal{L}_s$ and monotonicity term $\mathcal{L}_m$, are employed. In addition to these terms, we employ an LPIPS loss [40] function that assesses a solution concerning perceptually relevant characteristics (*e.g.*, the structural contents and detailed textures):

$$\mathcal{L}_p = \sum_l \frac{1}{H^l W^l} \sum_{h,w} \| \phi(\hat{\mathbf{I}})_{hw}^l - \phi(\mathbf{R})_{hw}^l \|_2^2, \tag{7}$$

where $\phi(\cdot)_{hw}^l$ denotes the feature map of layer $l$ extracted from a pre-trained AlexNet [20].

To summarize, the complete objective of our proposed model is combined as follows:

$$\mathcal{L} = \mathcal{L}_1 + \lambda_s \mathcal{L}_s + \lambda_m \mathcal{L}_m + \lambda_p \mathcal{L}_p, \tag{8}$$

where $\lambda_s$, $\lambda_m$, and $\lambda_p$ are hyper-parameters to control the balance of loss functions. In our experiment, these parameters are set to $\lambda_s = 0.0001$, $\lambda_m = 10$, $\lambda_p = 0.01$.

| Methods | #Params | MIT-FiveK (480p) | | | | MIT-FiveK (original) | | | |
|---|---|---|---|---|---|---|---|---|---|
| | | PSNR↑ | SSIM↑ | LPIPS↓ | $\triangle E$↓ | PSNR↑ | SSIM↑ | LPIPS↓ | $\triangle E$↓ |
| UPE [35] | 999K | 21.82 | 0.839 | 0.136 | 9.16 | 20.41 | 0.789 | 0.253 | 10.81 |
| HDRNet [12] | 482K | 23.31 | 0.881 | 0.075 | 7.73 | 22.99 | 0.868 | 0.122 | 7.89 |
| CSRNet [14] | **37K** | 25.31 | 0.909 | **0.052** | 6.17 | 24.23 | 0.891 | 0.099 | 7.10 |
| DeepLPF [26] | 1.72M | 24.97 | 0.897 | 0.061 | 6.22 | N.A. | N.A. | N.A. | N.A. |
| LUT [38] | 592K | 25.10 | 0.902 | 0.059 | 6.10 | 23.27 | 0.876 | 0.111 | 7.39 |
| sLUT [36] | 4.52M | 24.67 | 0.896 | 0.059 | 6.39 | 24.27 | 0.876 | 0.103 | 6.59 |
| CLUT [39] | 952K | 24.94 | 0.898 | 0.058 | 6.71 | 23.99 | 0.874 | 0.106 | 7.07 |
| **Ours** | 731K | **25.53** | **0.910** | 0.055 | **5.64** | **24.52** | **0.897** | **0.081** | **6.34** |

Table 2: Quantitative comparison on MIT-Adobe FiveK [4] dataset. "N.A." means that the results are not available due to insufficient memory of the GPU.

## 3 Experiments

### 3.1 Experimental Setup

**Datasets:** We evaluate the performance of our network on two challenging benchmark datasets: MIT-Adobe FiveK [4] and HDR+ burst photography [13]. The MIT-Adobe FiveK dataset is widely recognized as a benchmark for evaluating photographic image adjustments. This dataset comprises 5000 raw images, each retouched by five professional photographers. In line with previous studies [38, 36, 39], we utilize the ExpertC images as the reference images and adopt the same data split, with 4500 image pairs allocated for training and 500 image pairs for testing purposes. The HDR+ dataset is a burst photography dataset collected by the Google camera group to research high dynamic range (HDR) and low-light imaging on mobile cameras. We post-process the aligned and merged frames (DNG images) into 16-bit TIF images as the input and adopt the manually tuned JPG images as the corresponding reference images. We conduct experiments on both the 480p resolution and 4K resolution. The aspect ratios of source images are mostly 4:3 or 3:4.

**Evaluation metrics:** We employ four commonly used metrics to quantitatively evaluate the enhancement performance on the datasets as mentioned above. The $\triangle E$ metric is defined based on the $L_2$ distance in the CIELAB color space. The PSNR and SSIM are calcuated by corresponding functions in *skimage.metrics* library and RGB color space. Note that higher PSNR/SSIM and lower LPIPS/$\triangle E$ indicate better performance.

**Implementation Details:** To optimize the network, we employ the Adam optimizer [18] for training. The initial values of the optimizer's parameters, $\beta_1$ and $\beta_2$, are set to 0.9 and 0.999, respectively. The initial learning rate is set to $2 \times 10^{-4}$, and we use a batch size of 1 during training. In order to augment the data, we perform horizontal and vertical flips. The training process consists of 200 epochs. The implementation is conducted on the Pytorch [31] framework with Nvidia Tesla V100 32GB GPUs.

### 3.2 Quantitative Comparison Results

In our evaluation, we comprehensively compare our proposed network with state-of-the-art learning-based methods for tone mapping in the camera-imaging pipeline. The methods included in the comparison are UPE [35], DeepLPF [26], HDRNet [12], CSRNet [14], 3DLUT [38], spatial-aware 3DLUT [36], and CLUT-Net [39]. To simplify the notation, we use the abbreviations LUT, sLUT, and CLUT to represent 3DLUT, spatial-aware 3DLUT, and CLUT-Net, respectively, in our comparisons. It is important to note that the input images considered in our evaluation are 16-bit uncompressed images in the CIE XYZ color space, while the reference images are 8-bit compressed images in the sRGB color space.

Among the considered methods, DeepLPF and CSRNet are pixel-level methods based on ResNet and U-Net backbone, while HDRNet and UPE belong to patch-level methods, and LUT, sLUT, and CLUT are the image-level methods. Our method also falls within the image-level category. These methods are trained using publicly available source codes with recommended configurations, except

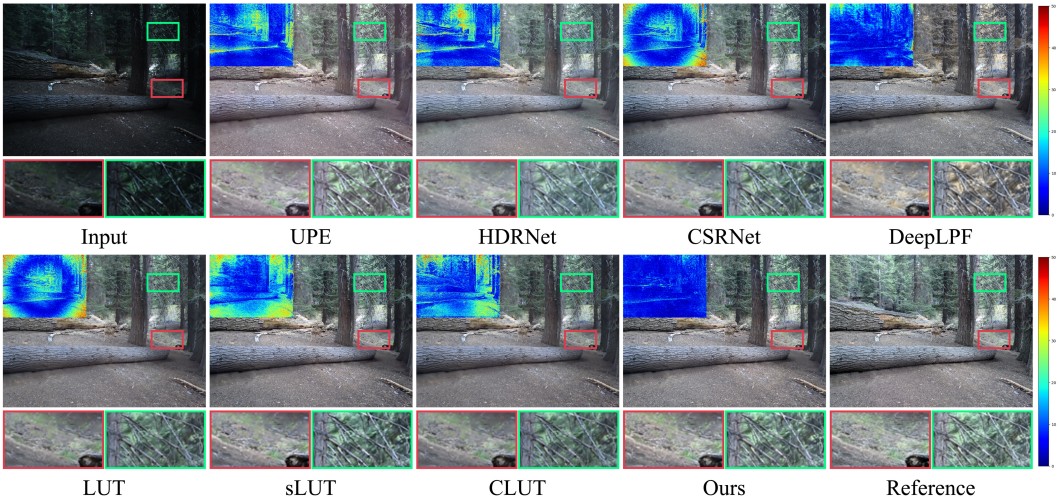

| Input | UPE | HDRNet | CSRNet | DeepLPF |

| LUT | sLUT | CLUT | Ours | Reference |

Figure 4: Visual comparison with state-of-the-art methods on a test image from the HDR+ dataset [13]. The error maps in the upper left corner facilitates a more precise determination of performance differences. Best viewed in color and by zooming in.

for sLUT. Since the training code and weights of this method have never been released, we reproduce the results according to the description in the released article.

Tab. 1 presents the quantitative comparison results on the HDR+ dataset for two different resolutions. Notably, our method exhibits a significant performance advantage over all competing methods on both resolutions, as indicated by the values highlighted in bold across all metrics. Specifically, our method achieves a notable 0.49dB improvement in PSNR compared to the second-best method, sLUT [36], at 480p resolution. This advantage becomes even more pronounced (1.25dB) when operating at the original image resolution, demonstrating the robustness of our approach for high-resolution images. Similarly, when evaluated on our second benchmark, the MIT-Adobe FiveK dataset (refer to Tab. 2), our method consistently demonstrates a clear advantage over all competing methods. However, for all methods, the FiveK dataset offers limited improvements compared to the HDR+ dataset, which can be attributed to two main reasons. Firstly, some reference images in the FiveK dataset suffer from overexposure or oversaturation, presenting challenges for the enhancement methods. Secondly, inconsistencies exist in the reference images adjusted by the same professional photographers, leading to variations between the training and test sets. More discussions can be found in the supplementary material.

### 3.3   Qualitative Comparison Results

To evaluate our proposed network intuitively, we visually compare enhanced images on the two benchmarks, as shown in Fig. 4 and Fig. 5. Note that the input images are 16-bit TIF images, which regular display devices can not directly visualize; thus, we compress the 16-bit images into 8-bit images for visualization. These figures show that our proposed network consistently delivers visually appealing results on the MIT-Adobe FiveK and HDR+ datasets. For example, in Fig. 4, our method excels in preserving intricate details such as tree branches and grass texture while enhancing brightness. Moreover, our results exhibit superior color fidelity and alignment with the reference image. Similarly, in Fig.5, while other methods suffer from poor saturation in the shaded area of the reflected building, our method accurately reproduces the correct colors, resulting in a visually pleasing outcome. These findings highlight the effectiveness and superiority of our method in tone mapping tasks. More visual results can be found in the supplementary material. Since the central goal of the tone mapping task is to primarily recalibrate the tone of the image while compressing the dynamic range, the visual differences between the results produced by the various state-of-the-art methods are minimal. To intuitively demonstrate the visual differences, we utilize the error maps to facilitate a more precise identification of performance differences.

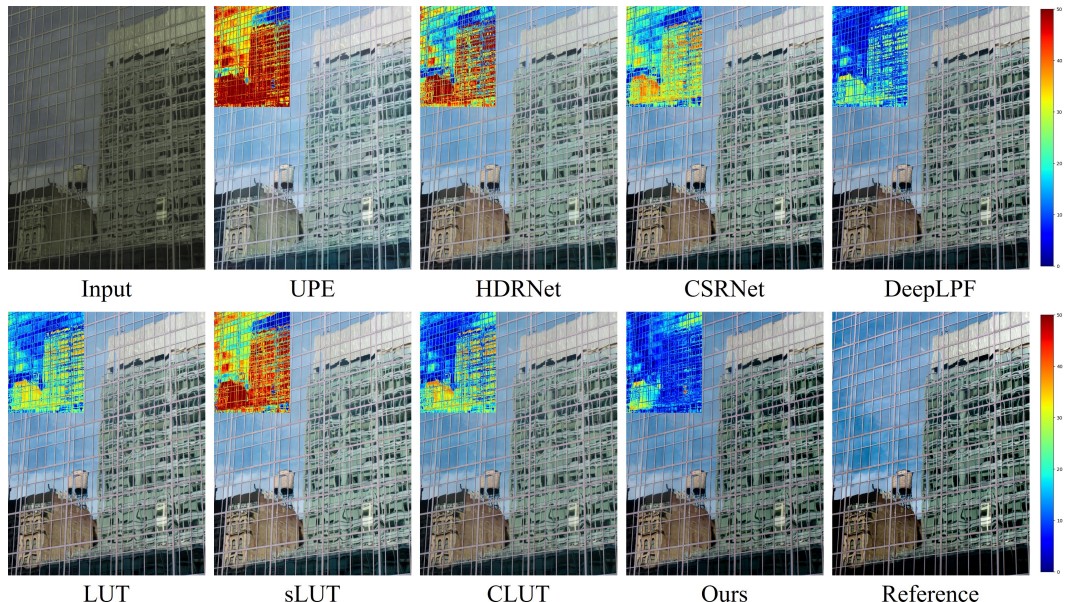

Figure 5: Visual comparison with state-of-the-art methods on a test image from the MIT-Adobe FiveK dataset [4]. The error maps in the upper left corner facilitates a more precise determination of performance differences. Best viewed in color and by zooming in.

| Metrics | Baseline | Framework Component | | | Low-frequency Image Resolution | | |
| | | + Weight Map | + Transformer | + Learnable Filter | 64 × 64 | 128 × 128 | 256 × 256 |
|---|---|---|---|---|---|---|---|
| PSNR↑ | 23.16 | 24.41 (+1.250) | 25.34 (+0.930) | 26.62 (+1.280) | 26.62 | 26.69 (+0.070) | 26.81 (+0.120) |
| SSIM↑ | 0.842 | 0.856 (+0.014) | 0.868 (+0.012) | 0.907 (+0.039) | 0.907 | 0.909 (+0.002) | 0.913 (+0.004) |
| LPIPS↓ | 0.113 | 0.111 (-0.002) | 0.101 (-0.010) | 0.063 (-0.038) | 0.063 | 0.061 (-0.002) | 0.058 (-0.003) |
| △E↓ | 7.04 | 6.23 (-0.81) | 5.92 (-0.31) | 5.31 (-0.61) | 5.31 | 5.29 (-0.02) | 5.25 (-0.04) |

Table 3: Ablation study of framework component and the selection of pyramid layers. All four metrics are reported.

## 3.4 Ablation Study

**Break-down Ablations.** We conduct comprehensive breakdown ablations to evaluate the effects of our proposed framework. We train our framework from scratch using paired data from the HDR+ dataset [13] and evaluate its performance on the HDR+ test set. The quantitative results are presented in Tab. 3. We begin with the baseline method, 3D LUT [38], without utilizing pixel-level weight maps or learnable local Laplacian filters. The results show a significant degradation, indicating the insufficiency of 3D LUT. When pixel-level weight maps are introduced, the results improve by an average of 1.25 dB. This evidence highlights the successful implementation of the pixel-level basis 3D LUTs fusion strategy discussed in Sec. 2.2. Next, we replace the regular lightweight CNN backbone with a tiny transformer backbone proposed by [23], which contains less than 400K parameters. After deploying the transformer backbone, the model is improved by 0.93dB, suggesting that the transformer backbone is more in line with global tone manipulation and benefits generating more visual pleasure LDR images. Furthermore, when employing the image-adaptive learnable local Laplacian filter, the results exhibit a significant improvement of 1.28 dB. This finding indicates that the image-adaptive learnable local Laplacian filter facilitates the production of more vibrant results. As can be seen from Fig 6, combining local Laplacian filter with 3D LUT achieves good visual quality on both global and local enhancement in this challenging case. These results convincingly demonstrate the superiority of our proposed framework in tone mapping tasks.

**Selection of the pyramid layers.** We validate the influence of the number of Laplacian pyramid layers in this section. Our approach employs an adaptive Laplacian pyramid, allowing us to manipulate the

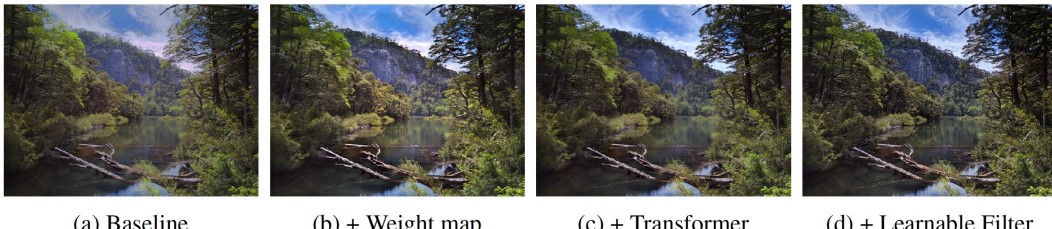

| (a) Baseline | (b) + Weight map | (c) + Transformer | (d) + Learnable Filter |

Figure 6: Visual results of ablation study on framework component. (a) is the baseline method 3D LUT. (b) is apply the pixel-level weight map. (c) is deploy the transformer backbone. (d) is utilizing the learnable local Laplacian filter.

number of layers by altering the resolution of the low-frequency image $I_{low}$. As shown in Tab. 3, the model performs best on all evaluation metrics when the resolution of $I_{low}$ is set to $256 \times 256$. However, the proposed framework requires more computation. A trade-off between computational load and performance is determined by the number of layers in the Laplacian pyramid. The proposed framework remains robust when the resolution is reduced to alleviate the computational burden. For example, reducing the resolution of $I_{low}$ from $256 \times 256$ to $64 \times 64$ only marginally decreases the PSNR of the proposed framework from 26.81 to 26.62. Remarkably, this reduction in resolution leads to a significant 30% decrease in computational burden. These results validate that the tone attributes are presented in a relatively low-dimensional space.

## 4 Conclusion

This paper proposes an effective end-to-end framework for HDR image tone mapping tasks, combining global and local enhancements. The proposed framework utilizes the Laplacian pyramid decomposition technique to handle high-resolution HDR images effectively. This approach significantly reduces computational complexity while simultaneously ensuring uncompromised enhancement performance. Global tone manipulation is performed on the low-frequency image using 3D LUTs. An image-adaptive learnable local Laplacian filter is proposed to progressively refine the high-frequency components, preserving local edge details and reconstructing the pyramids. Extensive experimental results on two publically available benchmark datasets show that our model performs favorably against state-of-the-art methods for both 480p and 4K resolutions.

**Acknowledgements.** This work was supported by the National Natural Science Foundation of China No.62176097, Hubei Provincial Natural Science Foundation of China No.2022CFA055.

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
