# The Supplementary Materials of the Main Paper
# Lookup Table meets Local Laplacian Filter: Pyramid Reconstruction Network for Tone Mapping

**Feng Zhang** [1]     **Ming Tian** [1]     **Zhiqiang Li** [2]     **Bin Xu** [2]
**Qingbo Lu** [2]     **Changxin Gao** [1,3*]     **Nong Sang** [1]

[1] National Key Laboratory of Multispectral Information Intelligent Processing Technology,
School of Artificial Intelligence and Automation, Huazhong University of Science and Technology,
[2] DJI Technology Co., Ltd, [3] Hubei Key Laboratory of Brain-inspired Intelligent Systems
{fengzhangaia, tianming, cgao, nsang}@hust.edu.cn,
{mila.xu, cristopher.li, qingbo.lu}@dji.com

This document shares more backgrounds, motivations, discussions, qualitative results, broader impacts, limitations, and future works not included in our main paper due to space limitations.

## 1   Background

### 1.1   Background of 3-Dimentional Look-up Table

A three-Dimensional Look-up Table (3D LUT) is often used to transform one set of colors into another. Given a source color $\{\mathbf{I}_{i,j,k}^r, \mathbf{I}_{i,j,k}^g, \mathbf{I}_{i,j,k}^b\}$, where $i, j, k = 0, 1, \cdots, n_b - 1 \in \mathbb{I}_0^{N_b-1}$, we want to find a destination tuned color $\{\mathbf{O}_{i,j,k}^r, \mathbf{O}_{i,j,k}^g, \mathbf{O}_{i,j,k}^b\}$. The precision of this transformation is determined by the number of elements $N_b$, where $N_b$ is the number of bins in each color channel and is typically set to 33 in practice. Since 3D LUT is a discrete function and color is continuous, it is typically necessary to interpolate between the nearest neighbors to calculate the output color. Assume that the 3D LUT is of size $N_b \times N_b \times N_b$. We can calculate the indices $\{i^r, i^g, i^b\}$ into the 3D LUT as follows:

$$i^r = \frac{\mathbf{I}_{i,j,k}^r}{\triangle}, \ i^g = \frac{\mathbf{I}_{i,j,k}^g}{\triangle}, \ i^b = \frac{\mathbf{I}_{i,j,k}^b}{\triangle} \tag{1}$$

where $\triangle = C_{max}/(N_b - 1)$, $C_{max}$ is the maximum color value.

Notice that we have yet to round these to integers. Instead, we get the integer and fractional parts of these indices separately:

$$\hat{i}^r = \lfloor i^r \rfloor, \ \hat{i}^g = \lfloor i^g \rfloor, \ \hat{i}^b = \lfloor i^b \rfloor \tag{2}$$

$$d^r = i^r - \hat{i}^r, \ d^g = i^g - \hat{i}^g, \ d^b = i^b - \hat{i}^b \tag{3}$$

where $\lfloor \cdot \rfloor$ is a floor function that rounds a number down to the nearest integer, the integer parts are indexed directly into the 3D LUT, and the fractional parts are used for interpolation.

After the location of the input RGB color is computed, its nearest eight surrounding elements can be used to interpolate the out RGB color via trilinear interpolation. The transformed output color

---

*Changxin Gao is the corresponding author, email: cgao@hust.edu.cn

37th Conference on Neural Information Processing Systems (NeurIPS 2023).

$\{\mathbf{O}^r_{i,j,k}, \mathbf{O}^g_{i,j,k}, \mathbf{O}^b_{i,j,k}\}$ can be derived by the trilinear interpolation as follows:

$$
\begin{aligned}
\mathbf{O}^c_{i,j,k} = & (1 - d^r)(1 - d^g)(1 - d^g)\mathbf{I}^c_{i,j,k} + d^r(1 - d^g)(1 - d^b)\mathbf{I}^c_{i+1,j,k} \\
& + (1 - d^r)d^g(1 - d^b)\mathbf{I}^c_{i,j+1,k} + (1 - d^r)(1 - d^g)d^b\mathbf{I}^c_{i,j,k+1} \\
& + d^r d^g(1 - d^b)\mathbf{I}^c_{i+1,j+1,k} + (1 - d^r)d^g d^b\mathbf{I}^c_{i,j+1,k+1} \\
& + d^r(1 - d^g)d^b\mathbf{I}^c_{i+1,j,k+1} + d^r d^g d^b\mathbf{I}^c_{i+1,j+1,k+1}
\end{aligned}
\tag{4}
$$

where $c \in \{r, g, b\}$. The transformation process using a 3D LUT is highly conducive to parallel computing architectures, such as the Graphics Processing Unit (GPU), primarily due to the independence of individual pixel computations from one another.

## 1.2 Background of Local Laplacian Filter

The Local Laplacian Filter is an image processing filter that combines the advantages of the Laplacian pyramid and bilateral filter for edge-preserving image smoothing and detail enhancement. It was first introduced by [9].

The basis of the filter is the Laplacian pyramid, a linear, multi-scale representation for images introduced by [2]. The Laplacian pyramid is formed by successively smoothing the original image with a Gaussian filter and subtracting adjacent levels of the smoothed image to form a set of band-pass images.

The Laplacian pyramid provides a way to represent the details at different scales in an image. However, it does not inherently provide a way to modify the details content-awarely. It is where the idea of the Local Laplacian Filter comes in. The filter combines the Laplacian pyramid with the concept of the bilateral filter [10], an edge-preserving smoothing filter. However, instead of applying the bilateral filter directly, which can lead to undesirable artifacts, it computes a set of modified Laplacian pyramids that emphasize or deemphasize details at different scales. The procedure of applying a Local Laplacian Filter can be summarized as follows:

**Laplacian Pyramid Construction.** Given an input image $\mathbf{I}$, we apply a Gaussian filter $G$ at level $\ell$ of the pyramid, resulting in a smoothed image $I_\ell$. The subsequent pyramid level is then produced by subsampling $I_\ell$ by a factor of two. This operation is iteratively performed to yield a sequence of reduced-size images, $\{\mathbf{I}_0, \mathbf{I}_1, \mathbf{I}_2, ..., \mathbf{I}_N\}$, where $\mathbf{I}_0$ is the original image and $N$ is the total number of levels in the Gaussian pyramid. The Laplacian pyramid is obtained by taking the difference between consecutive Gaussian pyramid levels. Given two consecutive levels, $\mathbf{I}_\ell$ and $\mathbf{I}_{\ell+1}$, the Laplacian component $L_\ell$ is computed by expanding $\mathbf{I}_{\ell+1}$ to the same size as $\mathbf{I}_\ell$ (using upsampling and interpolation) and then subtracting it from $\mathbf{I}_\ell$. Mathematically, this process is represented by the following equation:

$$L_\ell = \mathbf{I}_\ell - expand(\mathbf{I}_{\ell+1}) \tag{5}$$

where $expand(\cdot)$ represents the operation that includes upsampling the image to twice its size, followed by Gaussian filtering. This operation is carried out for all levels of the Gaussian pyramid, resulting in the Laplacian pyramid. The top level of the Laplacian pyramid is identical to the top level of the Gaussian pyramid.

**Remapping Function Application.** The remapping function is applied to the input image per pixel, producing a new image. This function aims to adjust the pixel intensities in a way that amplifies or diminishes certain details while preserving the overall structure of the image. Since the user provides a parameter $\sigma_r$, intensity variations smaller than $\sigma_r$ should be considered fine-scale details and more significant variations are edges. As a center point for this function, we use $g_0 = \mathbf{I}_0(x_0, y_0)$, representing the image intensity at the location and scale where we compute the output pyramid coefficient. Intuitively, pixels closer than $\sigma_r$ to $g_0$ should be processed as details, and those farther than $\sigma_r$ away should be processed as edges. Combing Equation 3 in the main paper, the output coefficient $(\ell, x, y)$ can be obtained as follows:

$$
\mathbf{r(i)} = \begin{cases} g_0 + sign(i - g_0)\sigma_r(|i - g_0|/\sigma_r)^\alpha & if\ i \leq \sigma_r \\ g_0 + sign(i - g_0)(\beta(|i - g_0| - \sigma_r) + \sigma_r) & if\ i > \sigma_r \end{cases}, \tag{6}
$$

**Reconstruct the remapped components.** The final step involves reconstructing the Laplacian pyramid from the remapped components. The final output image $O$ is obtained by collapsing the

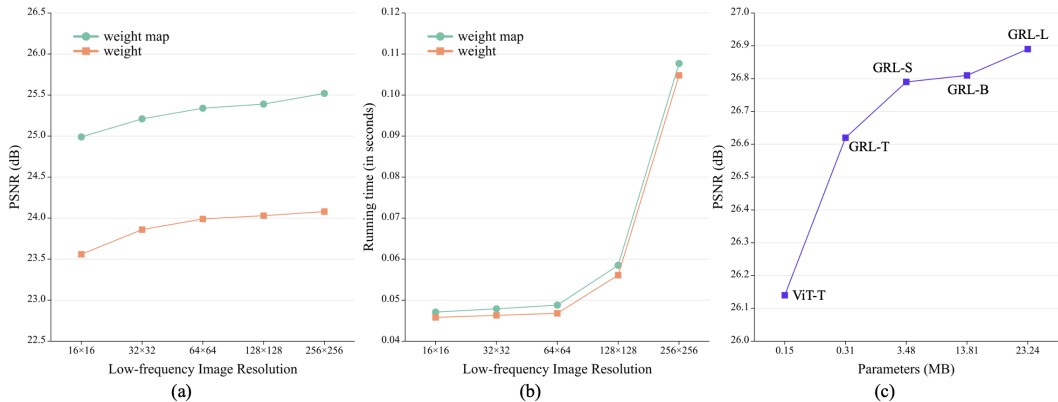

Figure 1: (a) shows the performance of the two fusion strategies on low-frequency images with different resolutions. (b) illustrates the difference in running speed of the two fusion strategies on low-frequency images with different resolutions. (c) presents the effect of models with different parametric quantities on the enhancement performance.

reconstructed Laplacian pyramid. This process involves expanding each pyramid level, adding it to the next higher level, and repeating this until the original resolution is achieved.

The Local Laplacian Filter effectively performs edge-preserving smoothing and detail enhancement in a content-aware manner, taking into account the local intensity structure of the image. It has applications in high dynamic range (HDR) compression, tone mapping, detail enhancement, and stylization.

### 1.3 Motivation of Using Local Laplacian Filter

As the limitation point out by [12], the 3D LUT remains uniform across different local regions in the image, which can result in suboptimal results in regions requiring local enhancement. This necessitates the incorporation of local enhancement techniques in order to achieve superior enhancement results.

Motivated by the work of [5], Zeng *et al.* [12] attempted to initially enhance the localization of the image by using a learnable guided filter, followed by a 3D LUT for global tone manipulation. Although this approach yields commendable results, it is still hampered by certain limitations. First, the combined approach requires two stages in the training process, precluding the possibility of end-to-end learning. Second, the learnable guided filter performs parameter learning on low-resolution images, thereby neglecting to take advantage of the high-frequency detail information inherent in the images.

To address these issues, we draw inspiration from the local Laplacian filter [9, 1] to amalgamate the effects of a local enhancement method with a 3D LUT through pyramid decomposition and reconstruction. This involves learning the filter parameters via a compact network at each layer and enhancing the Laplacian layer through a remapping function. This approach presents a potential solution to the previously described limitations, paving the way for more robust image enhancement methodologies.

## 2 Discussion

### 2.1 Discussion on basis 3D LUTs fusion strategy

Image-adaptive 3D LUT [12] presents an efficient approach to executing complex color transformations on digital media, including hue, saturation, and brightness adjustments. They propose to fuse the multiple basis 3D LUTs into an image content adaptive one using a soft-weighting strategy. Such a design improves the speed of processing high-resolution images. However, as described in Sec. 2.2 of the main paper, this strategy exhibits a drawback in accommodating complex pixel mapping relationships, thereby hindering its effectiveness in accurately representing such pixel transformations.

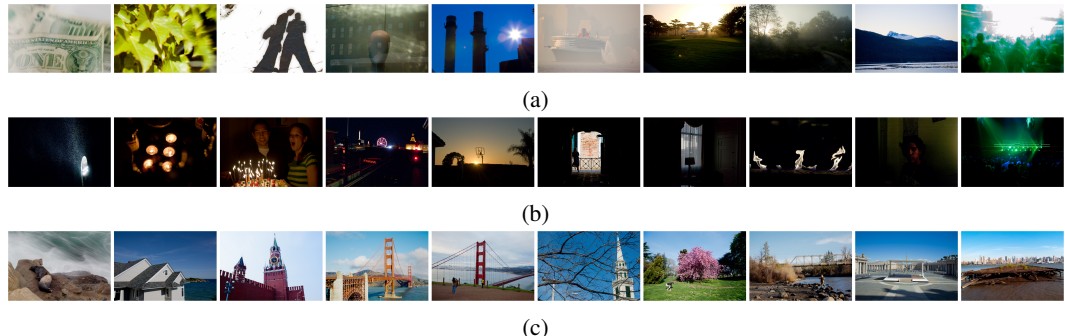

(a)

(b)

(c)

Figure 2: Example reference images of the MIT-Adobe FiveK dataset. (a) present the overexposure problems of retouched reference images; (b) the underexposure issues of retouched reference images; (c) the issue of inconsistent retouching style.

In order to rectify this inherent shortcoming, we propose an effective strategy for the fusion of basis 3D LUTs.

Although our proposed strategy enhances the capability of 3D LUT to handle complex pixel mapping relationships, it consequently leads to an increase in computational load due to the incorporation of pixel-level weight maps. As illustrated in Fig. 1, we compare the performance and running speed between our proposed strategy and the previous method. Specifically, at a resolution of $64 \times 64$ for the low-frequency image, our strategy demonstrates an improvement of 1.35 dB in performance while running merely 2 ms slower than the previous approach. Compared to a low-frequency image resolution of $64 \times 64$, the enhancement performance observed at a resolution of $256 \times 256$ pixels is relatively marginal, exhibiting an improvement of merely 0.18 dB. However, this minor performance gain is offset by a twofold increase in running time. When contrasting the results with a low-frequency image resolution of $16 \times 16$, we observe a performance enhancement of 0.35 dB at a resolution of $64 \times 64$. Interestingly, this improvement does not correspond with any notable increase in operation speed, suggesting a considerable boost in performance without sacrificing computational efficiency. Therefore, considering running speed and performance, we determine to execute global tone manipulation at the low-frequency image resolution of $64 \times 64$.

## 2.2 Discussion on Transformer Backbone

In this subsection, we have extended our analysis to examine the influence of the transformer backbone on the framework's performance. As presented in Fig. 1, a substitution of the GRL [7] model with the tiny ViT [4] model (ViT-T) yields a performance enhancement that decreases by a magnitude of 0.48 dB, concurrently, the quantity of model parameters is reduced by 160K. In the event that the GRL model is employed with a greater volume of parameters, there is a correlating increase in performance enhancement following the escalation in parameter quantity. However, this marginal performance improvement becomes relatively insignificant when compared to the large increase in the number of parameters. After critically evaluating the trade-off between the number of model parameters and the enhancement performance, we chose to utilize the tiny GRL model as the transformer weight predictor for our framework.

## 2.3 Discussion on MIT-Adobe FiveK Dataset

In Sec. 3.2, we briefly analyzed the reasons for the lack of substantial improvement of all methods on the MIT-Adobe FiveK citebychkovsky2011learning dataset. In this subsection, we will elaborate on this topic, providing a more comprehensive overview of the relevant details.

Fig. 2 illustrates certain issues inherent in the MIT-Adobe FiveK dataset. As demonstrated in the first row, many reference images suffer from problems related to overexposure after being retouched by Expert C. In contrast, the second row reveals an underexposure issue in the reference image. The third row presents an inconsistent retouching style, even within the tone of reference images that the same expert adjusts. These complications contribute to the inability of learning-based approaches to achieve satisfactory convergence and thus to improve enhancement performance effectively. The

| Methods | #Params | MACs | Runtime |
|---------|---------|------|---------|
| UPE | 999K | 1.146G | 8.42ms |
| CSRNet | 37K | 52.843G | 48.28ms |
| DeepLPF | 1.72M | 454.422G | 386.62ms |
| LUT | 592K | 0.676G | 2.67ms |
| sLUT | 4.52M | 20.316G | 9.63ms |
| CLUT | 952K | 9.391G | 8.35ms |
| UTMO | 4.45M | 1410.048G | 162.23ms |
| Ours | 731K | 2.923G | 20.51ms |

Table 1: Running time (in millisecond) comparison between our approach and state-of-the-art methods on resolution $3840 \times 2160 \times 3$. All methods are tested on NVIDIA V100 GPU.

prevalence of these problems may additionally be attributed to the historical context of the dataset, which was collected in 2011. At this juncture in time, learning-based methodologies were still in their infancy of development. Thus the dataset was not explicitly curated to train deep learning models. This lack of specificity in data collection could potentially contribute to the challenges encountered in model performance and convergence. Given this, we expect more research to focus on capturing more robust and diverse datasets for training deep learning models. It will help facilitate better performance in future learning-based tone mapping studies.

## 2.4 Runtime Analyse

To better characterize our model's computational complexity and efficiency, we provide a more diverse set of metrics, including the number of parameters, the multiply-accumulate operations (MACs), and the runtime. The results are shown in Table. 1.

All presented results are obtained using a 32GB NVIDIA V100 GPU. Runtime values are averaged over 1000 images, each with dimensions of $3840 \times 2160 \times 3$. Similarly, Multiply-Accumulate Operations (MACs) are computed based on the input dimensions $3840 \times 2160 \times 3$.

The results show that the extensive computational demands of CSRNet, DeepLPF, and UTMO [11] lead to slow processing. These results can be attributed to the fact that they are pixel-level methods, rendering them more reliant on hardware capabilities, especially when processing large-resolution images. Conversely, our approach makes a trade-off between computational complexity and performance.

## 2.5 More Comparison Results on Various Evaluation Metrics

To better demonstrate the superiority of our methods, we also provide the quantitative comparison results of all methods on TMQI, Fidelity, and Naturalness metrics in the Table. 2.

All these results are evaluated on the HDR+ 480p dataset. The Table. 2 shows that TMQI produces similar results for HLD [8], UTMO [11], and our approach. However, as shown in Fig. 3, visual results show substantial differences, emphasizing the need for a comprehensive assessment combining all metrics to express performance differences.

## 3 More Quanlitative Results

### 3.1 Visual Comparisons on 480p resolution

In this subsection, we have chosen to utilize the 480p HDR+ [6] and MIT-Adobe FiveK [3] datasets as released by [12], with no modifications implemented. More visual comparisons are provided in Fig. 4 and Fig. 5. Each figure presents the results derived from five methods, accompanied by their corresponding error maps, thereby providing a detailed comparison of the performances of each method.

| Methods | PSNR | SSIM | LPIPS | △ E | TMQI | Fidelity | Naturalness |
|---|---|---|---|---|---|---|---|
| UPE | 23.33 | 0.852 | 0.150 | 7.68 | 0.8789 | 0.8951 | 0.4213 |
| HDRNet | 24.15 | 0.845 | 0.110 | 7.15 | 0.8768 | 0.8968 | 0.4212 |
| CSRNet | 23.72 | 0.864 | 0.104 | 6.67 | 0.8880 | 0.8908 | 0.4529 |
| DeepLPF | 25.73 | 0.902 | 0.073 | 6.05 | 0.8752 | 0.8820 | 0.4419 |
| LUT | 23.29 | 0.855 | 0.117 | 7.16 | 0.8818 | 0.9018 | 0.4119 |
| sLUT | 26.13 | 0.901 | 0.069 | 5.34 | 0.8854 | 0.9017 | 0.4486 |
| CLUT | 26.05 | 0.892 | 0.088 | 5.57 | 0.8863 | 0.9020 | 0.4705 |
| HLD | 18.52 | 0.799 | 0.168 | 12.20 | 0.8862 | 0.8975 | 0.5092 |
| UTMO | 16.21 | 0.709 | 0.215 | 17.60 | 0.8893 | 0.9061 | 0.4723 |
| Ours | 26.62 | 0.907 | 0.063 | 5.31 | 0.8977 | 0.9089 | 0.5134 |

Table 2: More quantitative comparison results of all methods on TMQI, Fidelity, and Naturalness metrics. All these results are evaluated on the HDR+ 480p dataset.

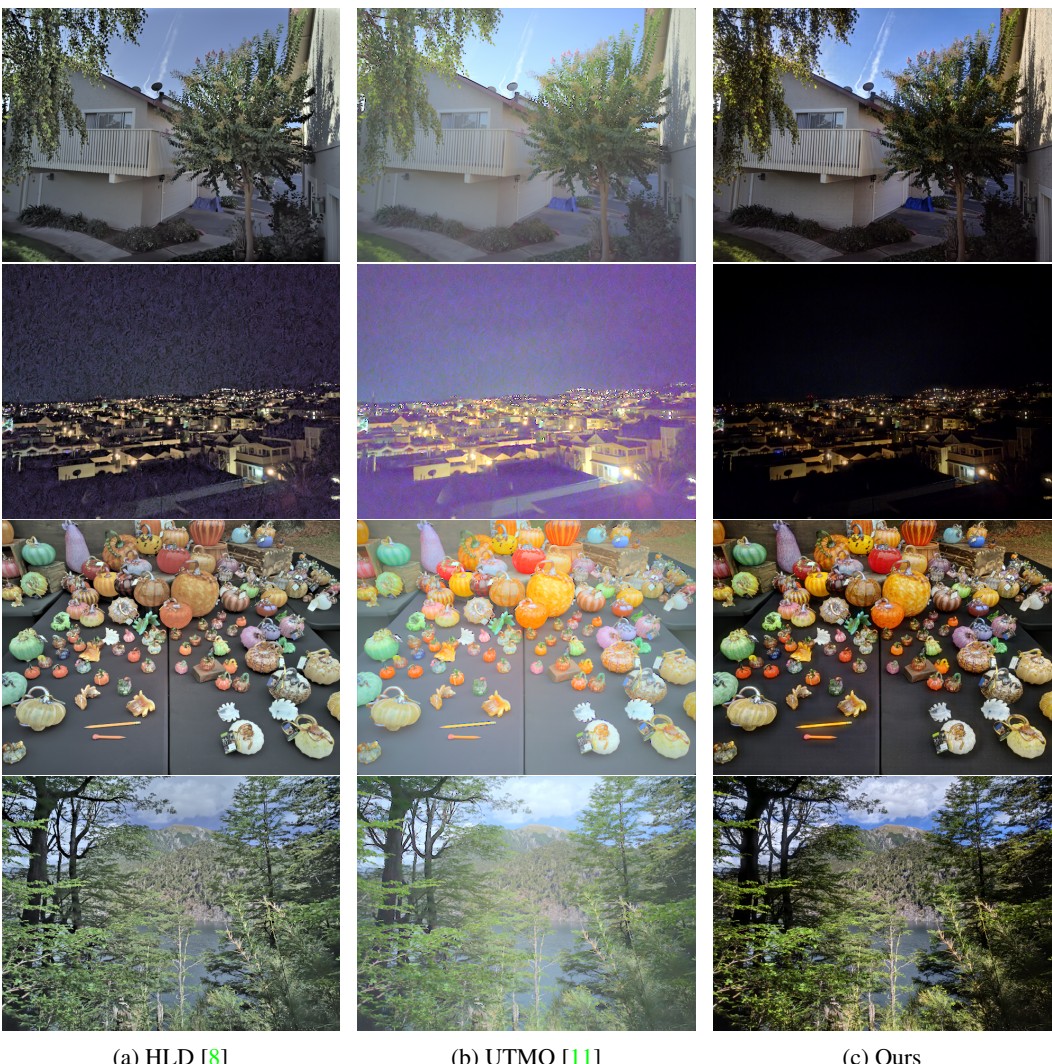

(a) HLD [8]          (b) UTMO [11]          (c) Ours

Figure 3: Visual comparison with method HLD [8] and UTMO [11] on test images from the HDR+ 480p dataset. Best viewed in color and by zooming in.

### 3.2 Visual Comparisons on 4K resolution

The proposed method, along with other state-of-the-art methods, has been trained and evaluated on the full-resolution (4K) HDR+ [6] and MIT-Adobe FiveK [3] datasets. A more exhaustive set of visual comparisons, serving to highlight the relative performances of each method, is available for scrutiny in Fig. 6 and Fig. 7.

## 4 Broader Impact

Tone Mapping is a fundamental task in the camera imaging pipeline. It has been studied for decades and is widely applied to cinematic color grading, computer graphics, and image processing. High dynamic range (HDR) imaging technology is gradually becoming ubiquitous as advancements are made in shooting equipment, rendering methods, and display devices. Hence, tone mapping techniques tackling this issue are worth investigating. The proposed framework addresses the lack of local edge detail prevalent in learning-based tone mapping methods.

The learning-based tone mapping methods mimic the retouching style of human photographers based on learned statistics of the training datasets. Consequently, these approaches may reflect any biases in the datasets, including those with negative social implications. The proposed framework presents this negative foreseeable societal consequence, either. Moreover, the subjective nature of tone mapping tasks means color transformations may not universally satisfy all users. A plausible mitigation strategy is to customize fine-tuning to various user preferences.

## 5 Limitation

Although our model demonstrates commendable efficacy concerning memory utilization and enhancement performance, it exhibits slower running time relative to the baseline method 3D LUT due to the constraints of the local Laplacian filter. However, this is a common limitation in local tone mapping operators. Fundamentally, this is because local operators are inherently more complex than global operators, and their effects vary on a pixel-by-pixel basis, depending on the local image characteristics. Nevertheless, since human visual perception is primarily sensitive to local contrast, local operators can provide superior performance. Due to its high degree of data parallelism, the local Laplacian filter [9, 1] can efficiently leverage multi-core architectures. Using OpenMP, we have achieved substantial GPU acceleration, thereby reducing runtime to the point where 4K image processing can be accommodated. This represents a significant advancement in addressing the running speed as mentioned above constraints and improving the overall performance of the model.

## 6 Future Work

The future works include: (**i**) Generalizing the framework to other computer vision applications, *e.g.*, low-light image enhancement, and style transfer. (**ii**) Modifying the architecture of the transformer backbone. (**iii**) Designing a customized fine-tuned strategy to satisfy various user preferences. (**iv**) Collecting a large-scale dataset for tone mapping benchmarking and researching.

Specifically, for future work (**i**), we generalize our proposed framework to other tasks since the conventional local Laplacian filter [9, 1] can be employed to many computer vision tasks. For future work (**ii**), the transformer backbone is directly borrowed from existing methods and is not optimized for the tone mapping tasks, which will be considered subsequently to improve the backbone structure. For future work (**iii**), we consider collecting mini-datasets with different user preferences, then feeding the cumulative histogram statistics of the datasets into the network as prior information, and fine-tuning the pre-trained model to achieve different retouching styles. This idea sounds reasonable and interesting. However, confirmation of its effectiveness by rigorous experiments is yet to be performed. Therefore, we set this task as a future research direction. We hope the present study will further stimulate scholars to explore this under-explored area of tone mapping.

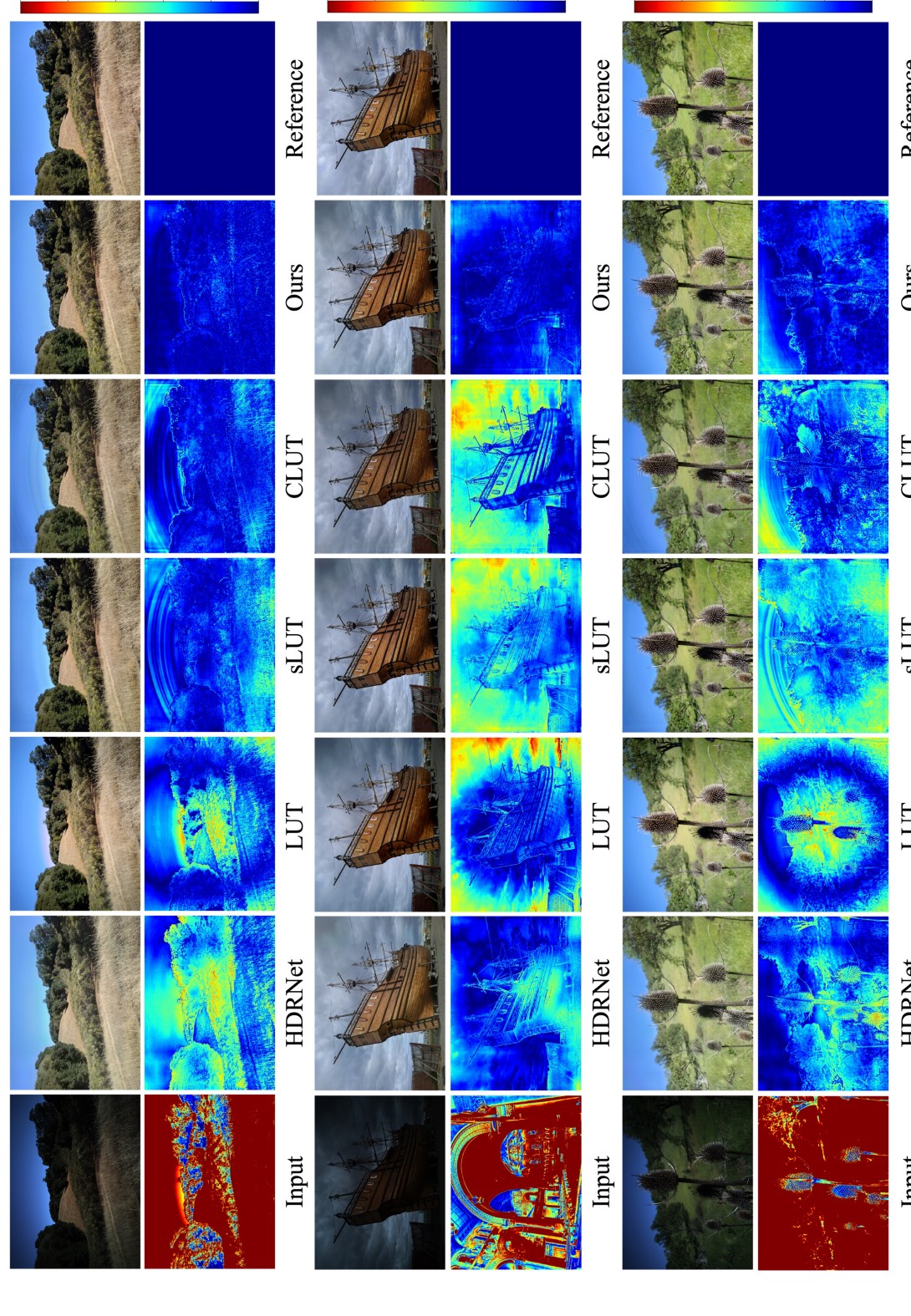

Figure 4: Visual comparisons on 480p resolution HDR+ dataset, and corresponding error maps. Best viewed in color and by zooming in.

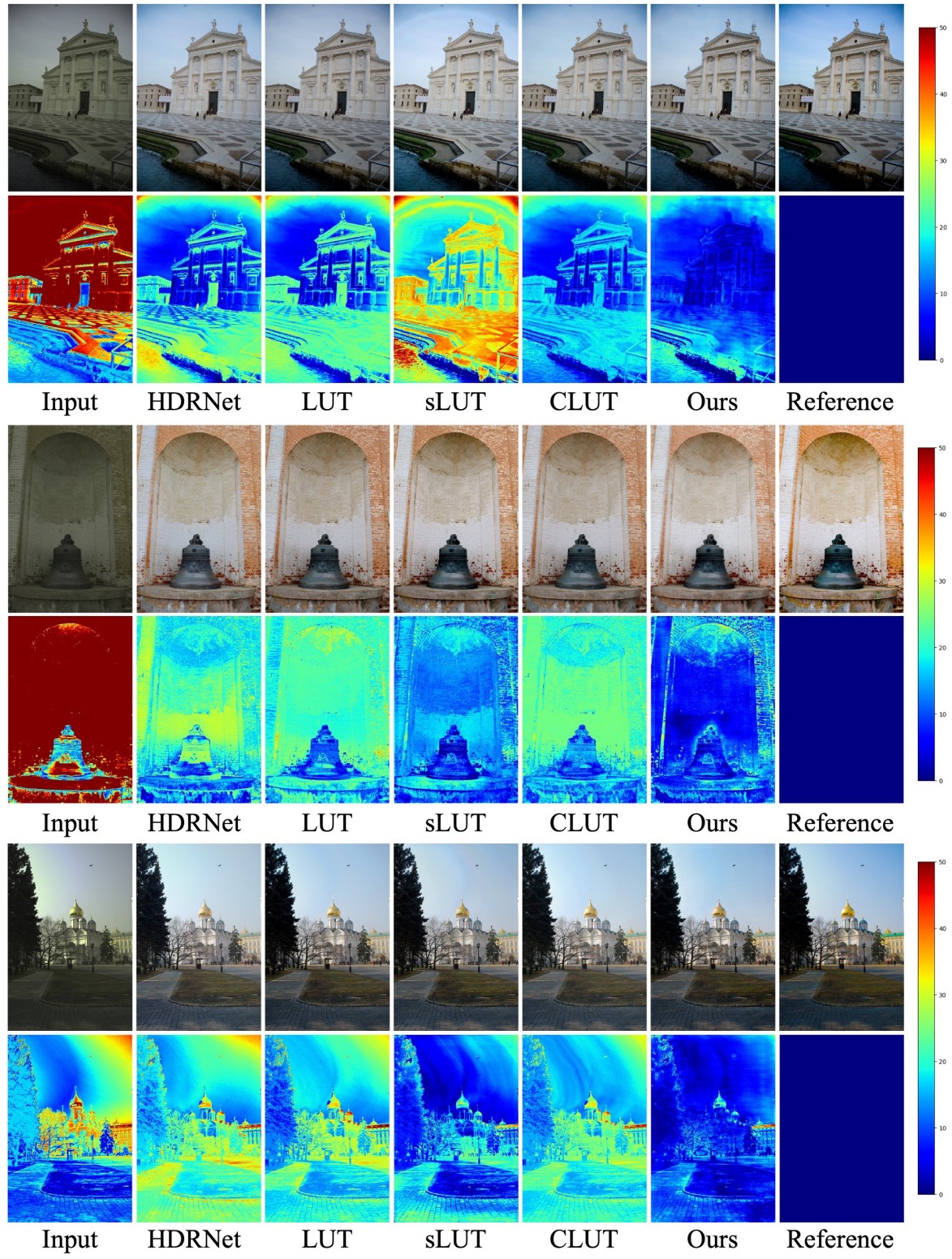

Figure 5: Visual comparisons on 480p resolution MIT-Adobe FiveK dataset, and corresponding error maps. Best viewed in color and by zooming in.

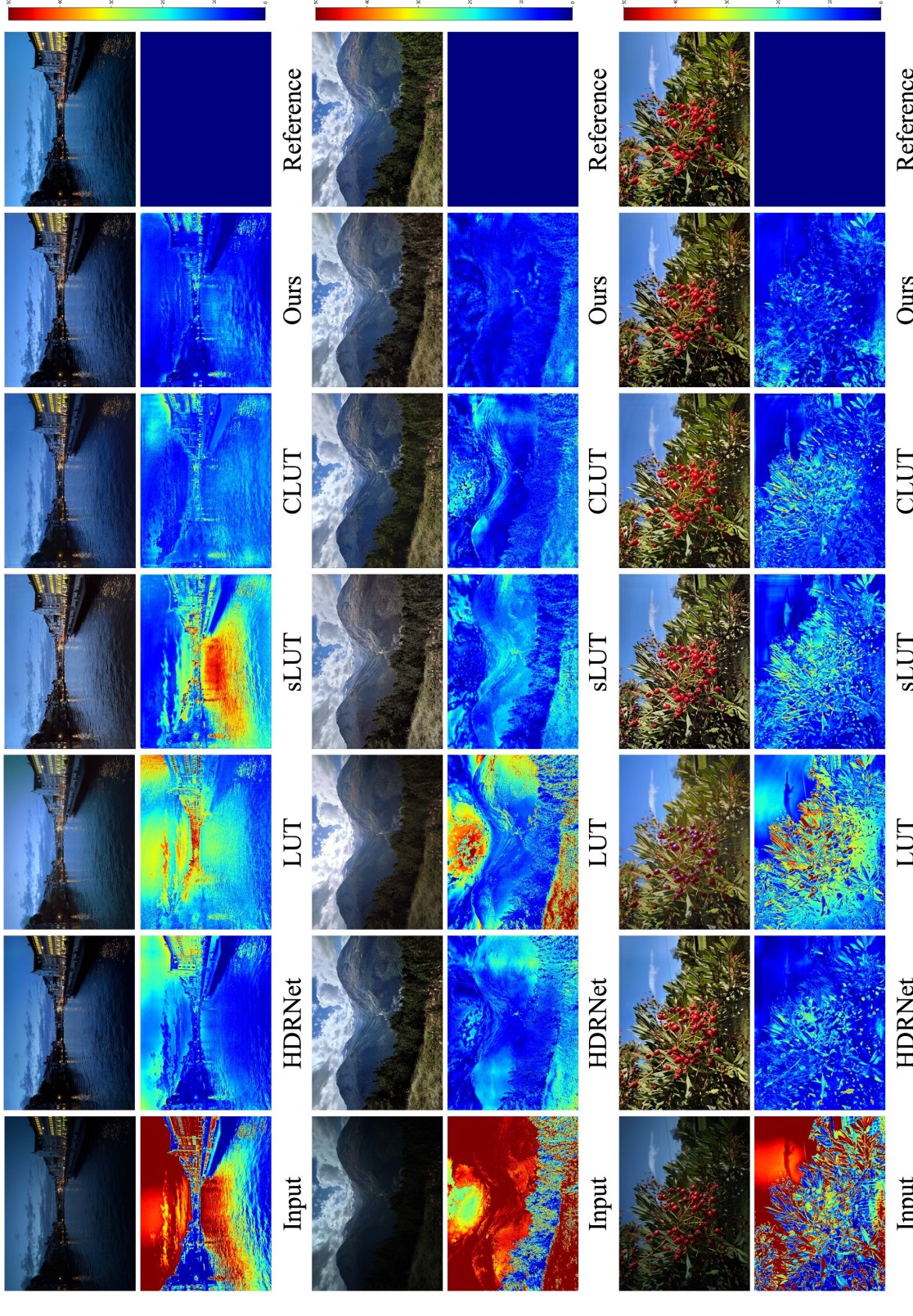

Figure 6: Visual comparisons on 4K resolution HDR+ dataset, and corresponding error maps. Best viewed in color and by zooming in.

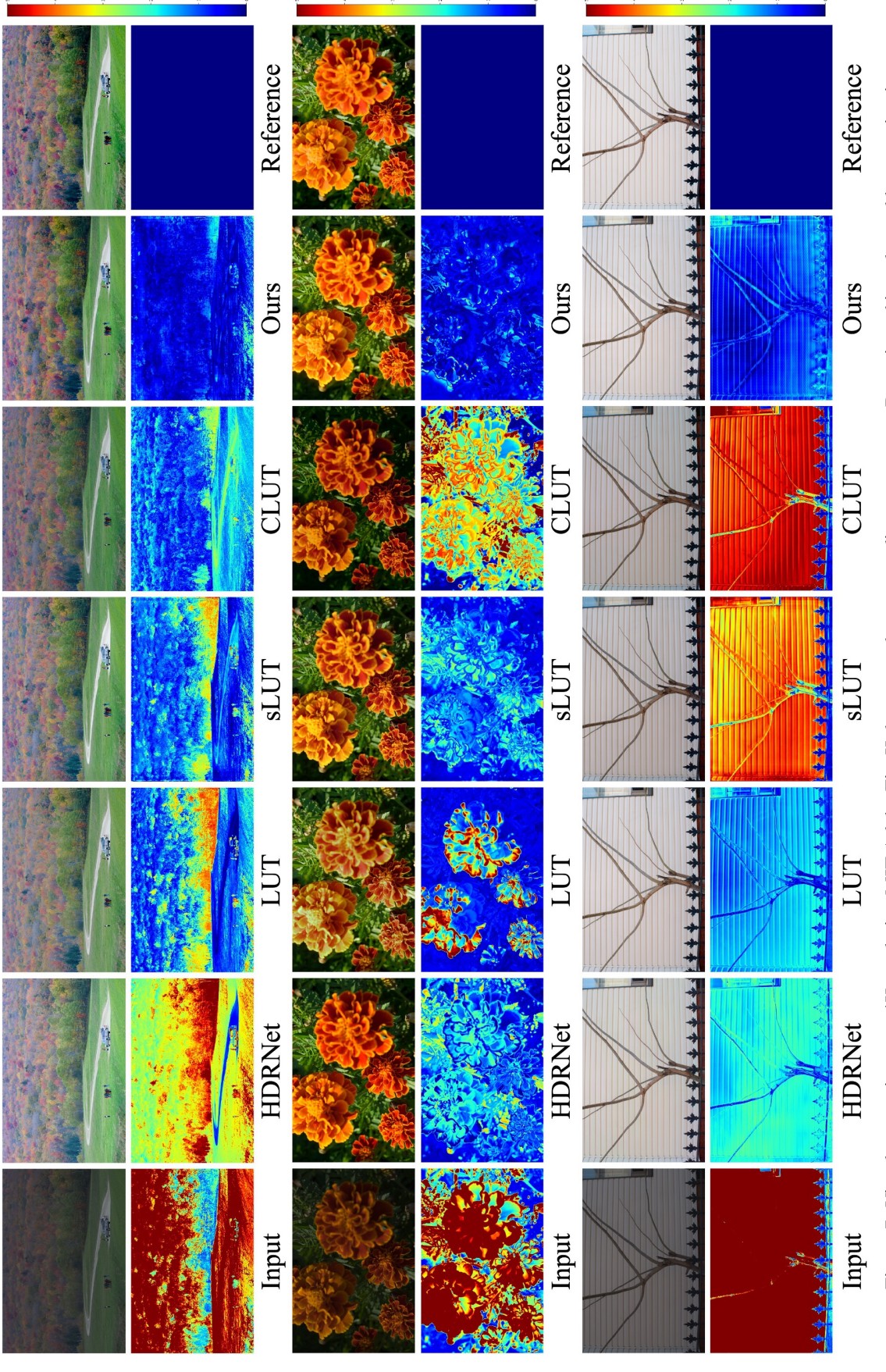

Figure 7: Visual comparisons on 4K resolution MIT-Adobe FiveK dataset, and corresponding error maps. Best viewed in color and by zooming in.