# OpenReview forum: "Lookup Table meets Local Laplacian Filter: Pyramid Reconstruction Network for Tone Mapping"
_NeurIPS.cc/2023/Conference — NeurIPS 2023 poster_

### Official Review · Reviewer_h55c · 2023-06-20

**Soundness:** 3 good
**Presentation:** 3 good
**Contribution:** 3 good
**Rating:** 6
**Confidence:** 4

**Summary:**

In this paper, the authors propose an effective end-to-end framework for HDR image tone mapping tasks, combining global and local enhancements. The proposed framework utilizes the Laplacian pyramid decomposition technique to handle high-resolution HDR images effectively. This approach significantly reduces computational complexity while simultaneously ensuring uncompromised enhancement performance. Global tone manipulation is performed on the low-frequency image using 3D LUTs. An image-adaptive learnable local Laplacian filter is proposed to progressively refine the high-frequency components, preserving local edge details and reconstructing the pyramids. Extensive experimental results on two publically available benchmark datasets show that our model performs favorably against state-of-the-art methods for both 480p and 4K resolutions.
The major contribution of this work is combination of high frequency lap filter and low frequency global tone mapping for acheiving new SOTA results for tone mapping.

**Strengths:**

New SOTA results for tone mapping using local learnable lap filters for refining high frequency details.

**Weaknesses:**

Generalization to other datasets may need retraining.

**Questions:**

How about mid frequency treatment ?


**Limitations:**

Improvement may be limited to scenes with complex lightings.

---

> ### Author Rebuttal · Authors · 2023-08-06
>
> Thanks for your valuable comments. Now we respond to your concerns one by one.
>
> (i) Generalization to other datasets may need retraining.
>
> Since tone mapping is a critical component of the camera imaging process, tailoring a model for each camera (and dataset) makes more sense. The subjective essence intrinsic to tone mapping tasks implies that universally applicable tone transformations may not uniformly meet the preferences of all users.
>
> In this regard, our future work will include the collection of diverse datasets, each containing different user preferences. Subsequently, the cumulative histogram statistics of these datasets will be integrated into the network as prior information. Subsequent steps include fine-tuning the pre-trained models to facilitate the realization of different tone mapping styles that accommodate the dynamic range of user preferences.
>
> (ii) Mid-frequency treatment.
>
> To the best of our knowledge, the Laplace pyramid contains only low-frequency and high-frequency information, which has been confirmed in previous studies [1-5] and has become a consistent observation. According to Burt and Adelson [1], each pixel in the low-frequency image is averaged over adjacent pixels by means of an octave Gaussian filter, which reflects the global characteristics of the input HDR image, including color and illumination attributes. Meanwhile, other high-frequency components contain edge-detailed textures of the image. The tone-mapped images can be obtained by directly utilizing the enhanced low and high-frequency information for pyramid reconstruction.
>
> (iii) Limitations.
>
> In fact, we analyze the broader impacts and limitations of our work in Sec. 4 and Sec. 5 of the supplementary materials. Regrettably, limited by the constraints imposed on paper length, we apologize for placing these pivotal sections in the supplementary materials. We agree with your comment that our method can not guarantee enhanced performance under complex lighting conditions. This is a common problem widely existing in data-driven deep learning methods. To alleviate this issue, we will augment data samples under complex lighting to fine-tune the network in our future work.
>
> [1] Peter J Burt and Edward H Adelson. The laplacian pyramid as a compact image code. In Readings in computer vision, Elsevier, 1987.
>
> [2] Sylvain Paris, Samuel W Hasinoff, and Jan Kautz. Local laplacian filters: Edge-aware image processing with a laplacian pyramid. ACM Trans. Graph, 2011.
>
> [3] Mathieu Aubry, Sylvain Paris, Samuel W Hasinoff, Jan Kautz, and Frédo Durand. Fast local laplacian filters: Theory and applications. ACM Transactions on Graphics (TOG), 2014.
>
> [4] Lai W S, Huang J B, Ahuja N, et al. Deep laplacian pyramid networks for fast and accurate super-resolution. Proceedings of the IEEE conference on computer vision and pattern recognition. 2017.
>
> [5] Liang J, Zeng H, Zhang L. High-resolution photorealistic image translation in real-time: A laplacian pyramid translation network. Proceedings of the IEEE/CVF Conference on Computer Vision and Pattern Recognition. 2021.

---

### Official Review · Reviewer_zrx6 · 2023-07-02

**Soundness:** 2 fair
**Presentation:** 2 fair
**Contribution:** 2 fair
**Rating:** 7
**Confidence:** 5

**Summary:**

The paper introduces a novel framework for converting high dynamic range (HDR) images to low dynamic range (LDR) images, a crucial task in digital imaging. The proposed framework aims to improve the balance between image enhancement and computational efficiency by incorporating global and local tone mapping operators.
Traditional methods are either global, processing the whole image uniformly but often missing fine details, or local, capturing fine details with higher computational requirements. To overcome this dichotomy, the authors propose integrating 3D LookUp Tables (3D LUT) as a global operator and local Laplacian filters as a local operator into one end-to-end system. The 3D LUTs are used for global tone manipulation, while local Laplacian filters refine high-frequency edge details.
Moreover, the authors introduce a learnable local Laplacian filter (LLF) and a lightweight network that progressively learns parameter value maps from annotated data for the local Laplacian filters. This approach reduces manual tuning, which is a limitation of existing methods.
The results of the extensive experiments on two benchmark datasets demonstrate that the proposed framework outperforms state-of-the-art methods in terms of both image quality and computational efficiency. The paper contributes to digital imaging by offering an efficient solution to HDR image tone mapping while preserving important details.

**Strengths:**

The paper is well-written and structured. The authors clearly explain the problems with existing methods and why their proposed framework offers a solution. Concepts are clearly defined, and the methodology is explained in a detailed and straightforward manner. Diagrams, equations, and examples effectively support the reader's understanding.

**Weaknesses:**

- The paper misses the discussions and comparisons of many related methods [1-3]. These methods are all published in good venues within the past five years and thus deserve a fair discussion.
- Why the evaluation metrics do not contain TMQI, Fidelity, and Naturalness? These three evaluation metrics are common practice in TMO papers (see [1] for example). As tone mapping is a subjective task, using PSNR or SSIM to evaluate does not provide much information.
- Some of the compared methods, such as CLUT [39], are tacking low-light image enhancement. Why did the authors compare these methods? The input of tone-mapping is an HDR radiance map, while the input to low-light image enhancement is an already tone-mapped non-linear image. The comparison is unfair and could mislead future work.
- I strongly suggest the authors revisit the goal of their proposed method. Is it a TMO (tone mapping operator) or low-light image enhancement (LLIE)? I really don't see many TMO methods in the comparison section. If that is the case, I would suggest the authors rename the paper title to low-light image enhancement and change all the terms in the paper.
- If the proposed method indeed is a TMO, then the first step should be LDR to HDR as the input is a low-light image. Then the authors should also discuss the related work [4-6] and design choices for LDR2HDR.
- While the authors tested their framework on two benchmark datasets, it would have been beneficial to see the system evaluated on more diverse datasets, including real-world photographs, video frames, and even synthetic images. This could have offered a more comprehensive understanding of the method's robustness and generalizability.
- The paper lacks a complex analysis of the proposed framework. A comparison of the computational complexity and memory usage between the proposed model and existing methods would give a clearer picture of the performance trade-off.
- The authors have not thoroughly discussed the potential limitations of their proposed method. Understanding what the model struggles with or where it might fail is essential for future improvements and applications.

[1] Liang, Zhetong, et al. "A hybrid l1-l0 layer decomposition model for tone mapping." Proceedings of the IEEE conference on computer vision and pattern recognition. 2018.

[2] Su, Chien-Chuan, et al. "Explorable tone mapping operators." 2020 25th International Conference on Pattern Recognition (ICPR). IEEE, 2021.

[3] Vinker, Yael, Inbar Huberman-Spiegelglas, and Raanan Fattal. "Unpaired learning for high dynamic range image tone mapping." Proceedings of the IEEE/CVF International Conference on Computer Vision. 2021.

[4] Liu, Yu-Lun, et al. "Single-image HDR reconstruction by learning to reverse the camera pipeline." Proceedings of the IEEE/CVF Conference on Computer Vision and Pattern Recognition. 2020.

[5] Santos, Marcel Santana, Tsang Ing Ren, and Nima Khademi Kalantari. "Single image HDR reconstruction using a CNN with masked features and perceptual loss." arXiv preprint arXiv:2005.07335 (2020).

[6] Chen, Xiangyu, et al. "Hdrunet: Single image hdr reconstruction with denoising and dequantization." Proceedings of the IEEE/CVF Conference on Computer Vision and Pattern Recognition. 2021.

**Questions:**

See the section above.

**Limitations:**

The paper does not discuss the limitation of the proposed method. I believe the proposed method will not have a potential negative societal impact.

---

> ### Author Rebuttal · Authors · 2023-08-06
>
> We thank the reviewer for the critical comments and good suggestions. Your comments help us to improve the quality of the manuscript. Below we respond to the reviewer’s comments and explain the changes we incorporated into the manuscript.
>
> (i) We thank the reviewer for referring us to these three studies [1]-[3]. We made comparisons between our proposed method and the methods in these studies.
>
> [1] and [2] are conventional tone mapping algorithms. On the other hand, [3] and our approach are deep learning-based but differ notably in methodology. [3] employs a pixel-level neural network for HDR image tone mapping. However, its computational complexity at larger image resolutions limits its application to resource-limited edge devices.
>
> In contrast, our approach combines a 3D Look-Up Table (3D LUT) with a fast local Laplacian filter, granting it a processing speed advantage. Moreover, our method outperforms both [1] and [3] across all metrics. Qualitative comparisons are illustrated in Fig. 1-4 in our provided rebuttal PDF, while quantitative results are presented in the following table:
>
> | Methods | PSNR   | SSIM   | LPIPS  | $\triangle$ E | TMQI   | Fidelity | Naturalness |
> |---------|--------|--------|--------|--------------|--------|----------|-------------|
> | [1]     | 18.52  | 0.799  | 0.168  | 12.20        | 0.8862 | 0.8975   | 0.5092      |
> | [2]     | N.A.   | N.A.   | N.A.   | N.A.         | N.A.   | N.A.     | N.A.        |
> | [3]     | 16.21  | 0.709  | 0.215  | 17.60        | 0.8893 | 0.9061   | 0.4723      |
> | Ours    | 26.62  | 0.907  | 0.063  | 5.31         | 0.8977 | 0.9089   | 0.5134      |
>
> N.A. represents no public source code for [2] is accessible. Despite efforts to contact the authors for assistance, no response was received within the rebuttal period. Thus, reproducing their work was unfeasible at this stage. However, our commitment to reproducing [2] remains, and we intend to include comparative results in our final version.
>
> (ii) Evaluation metrics do not contain TMQI, Fidelity, and Naturalness.
>
> Our method was mainly developed considering the possible side effect of noise caused by tone mapping. Therefore, our manuscript only included PSNR, SSIM, LPIPS, and $\triangle$E. We thank the reviewer for suggesting that TMQI, fidelity, and naturalness should also be considered. Due to rebuttal space constraints, we summarize the comparison results of all methods in the global rebuttal comments. It can be observed that our proposed method also produced acceptable results for TMQI, fidelity, and naturalness.
>
> (iii) Issues regarding the method of comparison.
>
> The reasons why we compare these methods can be summarized as follows:
>
> (a) CLUT was initially designed for photo retouching rather than low-light image enhancement. In our study, we modify CLUT by transforming the original 8-bit tone-mapped image into a 16-bit HDR radiance map and retraining it. This adaptation broadens CLUT's utility beyond traditional use, including tone mapping.
>
> (b) Given our reliance on lookup tables (LUTs), we compare our approach with state-of-the-art (SOTA) methods using LUTs. We have adapted these LUT-based techniques in our study by converting their input images to 16-bit HDR radiance maps. These adjustments were made for a thorough and fair comparative assessment.
>
> (c) Deep learning-based Tone Mapping Operator (TMO) techniques have seen limited exploration, with only a few articles in reputable venues over the past five years. It's also worth noting that some image enhancement methods can be adapted for TMO with minor changes. Therefore, we reference the classical 3D LUT [7] approach and selectively compare it to deep learning-based methods in our study.
>
> From my personal standpoint, within the realm of deep learning-based image processing methods, task names align with the input image nature. When dealing with an 8-bit tone-mapped non-linear image, the task is image enhancement (photo retouching). Conversely, a 16-bit HDR radiance map as input defines the task as tone mapping (imaging pipeline enhancement). This versatility in deep learning allows for achieving image enhancement and tone mapping goals.
>
> (iv) The goal of our proposed method.
>
> Our method is unambiguously tailored for tone mapping, and the paper emphasizes that the input image to the model is a 16-bit HDR radiance map. Remarkably, we seamlessly incorporate a 3D LUT with a fast local Laplacian filter, both of which are classical techniques in the tone mapping field.
>
> (v) LDR to HDR reconstruction.
>
> Our model was critically evaluated using the widely recognized HDR+ and MIT-Adobe FiveK datasets, utilizing 16-bit HDR radiance maps as inputs. Given this, the requirement to convert a single Low Dynamic Range (LDR) image into HDR [4]-[6] seems unnecessary in this context.
>
> (vi) Evaluated on more diverse datasets.
>
> To the best of our knowledge, the HDR+ and MIT-Adobe FiveK datasets are widely used for tone mapping tasks, and they are real-world photographs. Moreover, [1]-[3] were also evaluated on these datasets. Therefore, there need to be more alternative datasets that can be used for evaluation. In future work, we intend to collect a new dataset to expand the range of evaluation options.
>
> (vii) Runtime analyses.
>
> Please refer to the response to the question (ii) of reviewer nHJa for more comprehensive results on computational complexity and efficiency.
>
> (viii) Limitation and potential negative societal impact.
>
> In fact, we analyze the broader impacts and limitations of our work in Sec. 4 and Sec. 5 of the supplementary materials. Regrettably, limited by the constraints imposed on paper length, we apologize for placing these pivotal sections in the supplementary materials.
>
> [7] Hui Zeng, et al. Learning image-adaptive 3d lookup tables for high performance photo enhancement in real-time. IEEE Transactions on Pattern Analysis and Machine Intelligence, 2020.

---

### Official Review · Reviewer_1rH8 · 2023-07-06

**Soundness:** 3 good
**Presentation:** 3 good
**Contribution:** 3 good
**Rating:** 7
**Confidence:** 5

**Summary:**

This paper introduces an effective end-to-end framework for HDR image tone mapping. The network performs both global tone manipulation and local edge details preservation within the same model. The proposed method learns an image-adaptive learnable local Laplacian filter for local edge details preservation.

**Strengths:**

1. This paper is well-organized and easy to read.
2. The experiment results show its effectiveness on two benchmarks, as well as the visual results.
3. The proposed learnable local Laplacian filter shows remarkable improvement when integrated with image-adaptive 3D LUT.

**Weaknesses:**

1. Only limited benchmarks are valued, unlike previous work like CLUT, evaluated on both FiveK, HDR+, and PPR10K, so more evaluation results are encouraged.
2. There is no inference time presented in this work, which is another key information for enhancement applications.

**Questions:**

1. Why are the key metrics of baselines reported in this paper significantly lower than those reported in their original paper? For example, the PSNR of CLUT[39] on the HDR+ dataset is 26.98dB, while this figure is only 26.05dB in your Table 1, the PSNR of sLUT [36]  on the HDR+ dataset is 26.94dB, while this figure is 26.13dB in your paper, all these figures are much higher than yours. The same issues occur in Table 2.

**Limitations:**

The limitation is discussed in Supp materials.

---

> ### Author Rebuttal · Authors · 2023-08-06
>
> Thanks for your valuable comments. Now we respond to your concerns one by one.
>
> (i) Evaluation on PPR10K dataset.
>
> The PPR10K dataset is designed explicitly for portrait photo retouching, which deviates from the domain of tone mapping datasets. Notably, the CLUT being compared in our study is subject to adaptation. In the original paper of CLUT, the input image consisted of a nonlinear image (8-bit) that had been tone-mapped. In contrast, in our study, we reconfigured it to include the HDR radiogram (16-bit) as input. Consequently, since the PPR10K dataset does not contain HDR images, our model was not evaluated on this dataset.
>
> Nonetheless, it is essential to emphasize that our model can be readily extended for image enhancement purposes. We intend to explore this aspect by evaluating the PPR10K dataset in future work. We apologize for any ambiguity in this regard and will give a more precise elucidation in the final version.
>
> (ii) Runtime analyses.
>
> Please refer to the response to the question (ii) of reviewer nHJa for more comprehensive results on computational complexity and efficiency.
>
> (iii) Metrics of baselines lower than original paper.
>
> Firstly, it is essential to acknowledge that our primary task is tone mapping. Consequently, the CLUT and sLUT referenced in our study differ from their original counterparts, mainly in input images. Notably, both CLUT and sLUT considered in our work employ an HDR radiance map (16-bit) as the input image, diverging from their original papers where an already tone-mapped non-linear image (8-bit) is utilized.
>
> Secondly, it is worth mentioning that the source code of sLUT is still not publicly available. Despite our efforts to reproduce its results, we are still trying to match the original paper's results. Notably, the original CLUT paper specifies the integration of CLUT into sLUT, thus adding to the complexity of reproducing the results of the original paper. This combination of factors led to our findings' inconsistency with the original article's results.
>
> Moreover, these results were meticulously reviewed and endorsed by the authors of CLUT before our submission.

---

> > ### Comment · Reviewer_1rH8 · 2023-08-14
> > **A raise of the final rating from 6 to 7.**
> >
> > My concerns were well addressed and I would love to see this manuscript on the accepted list.

---

### Official Review · Reviewer_g4yF · 2023-07-07

**Soundness:** 2 fair
**Presentation:** 3 good
**Contribution:** 2 fair
**Rating:** 3
**Confidence:** 4

**Summary:**

The paper proposes an algorithm for the HDR image tone mapping task. It is an end-to-end framework combining global and local enhancements. It uses the Laplacian pyramid decomposition technique. It achieves a balance between computational complexity and performance. Global tone manipulation is performed on low-frequency component. Experiment shows that it out-performs SOTA.

**Strengths:**

It achieves a balance between computational complexity and performance.
Experiment shows that it out-performs SOTA.
It is intuitive to do global tone manipulation on low-frequency image, while using image-adaptive learnable local Laplacian filter for high-frequency component.
Nice ablation studies are performed to quantify the improvements from different modules of the whole algorithm.

**Weaknesses:**

I wonder why the the transformer weight predictor and the basis 3D LUTs fusion block only exist in the coarsest level of the pyramid. Is there any way to integrate them into all levels?
If integrate them into all levels, how much additional computational burden will be introduced?

The quantitative comparison on HDR+ dataset shows that CSRNet and LUT are both smaller than the proposed algorithm. Is there any chance to compare with them with similar amount of computation? E.g. if the proposed algorithm is reduced to 37K in terms of number of parameters, will it still outperform CSRNet?

LUT will use heavy storage. How to compare the proposed algorithm with SOTA in terms of data storage? Number of parameters alone will not tell the whole picture of the efficiency, what is the real running time comparison?




**Questions:**

see weaknesses

**Limitations:**

did not talk about potential negative societal impact

---

> ### Author Rebuttal · Authors · 2023-08-06
>
> Thanks for your valuable comments. Now we respond to your concerns one by one.
>
> (i) Integrate transformer model and fusion block to all pyramid levels.
>
> Primarily, the architectural design of our model focuses on the strategic goal of achieving a minimum number of parameters and computational requirements. This deliberate emphasis on efficiency aims to facilitate the deployment of the model on edge devices. Consequently, our approach is centered on minimizing the number of convolutional layers, effectively streamlining the computational footprint of the model.
>
> Secondarily, the rationale for deploying a lightweight transformer model at the coarsest layer stems from the intrinsic nature of the base of the Laplace pyramid. The coarsest layer of the Laplace pyramid is the low-frequency (global) information of the image, and the transformer model collects the global information of the image better than the CNN model. Furthermore, the placement of the LUT fusion module in proximity to this layer is deliberate. The LUT is a global tone-mapping operator, rendering it adept at processing global information. At the same time, the fast processing capability of the LUT mitigates the slower execution of the transformer model, thus effectively compensating for this temporal trade-off.
>
> (ii) Reduce the number of parameters and compare with CSRNet and LUT.
>
> The number of parameters in our model is distributed in three different parts. The first part consists of the transformer model, which requires 329K parameters; the second part consists of the LUT, which requires 315K parameters; and the last part is the compact network that predicts the parameters of the Laplace filter, which requires 87K parameters. The transformer model itself is characterized by a large number of parameters, so it becomes a challenge to reduce the number of parameters to 37K. To achieve this goal, we adopted a strategic approach of replacing the transformer model with a conventional CNN model while removing the LUT component and reducing the number of parameters of the compact network. As a result of the concerted effort, a target parameter count of 37K was finally achieved. Subsequently, the generated model was juxtaposed with CSRNet and LUT, and the results for the HDR+ dataset (480p) are fully presented in the following table:
>
> | Methods      | $\#$Params | PSNR  | SSIM  | LPIPS | $\triangle$ E |
> |--------------|------------|-------|-------|-------|--------------|
> | CSRNet       | 37K        | 23.72 | 0.864 | 0.104 | 6.67         |
> | Ours (37)    | 37K        | 23.91 | 0.872 | 0.098 | 6.32         |
> | LUT          | 592K       | 23.29 | 0.855 | 0.117 | 7.16         |
> | Ours (592)   | 592K       | 25.53 | 0.891 | 0.077 | 5.84         |
>
> (iii) The data storage of LUT.
>
> The data storage requirement for a Look-Up Table (LUT) is notably compact. For instance, the LUT employed in our study comprises dimensions of 33x33x33, culminating in a mere storage demand of 3x33x33x33 bits, translating to 107,811 bits, or approximately 0.1 megabytes (MB). Notably, our study incorporates three LUTs, which merge to form a singular LUT. Consequently, our study's cumulative storage allocation for all LUTs is a mere 0.3 MB.
>
> (iv) Runtime analyses.
>
> Please refer to the response to the question (\romannumeral2) of reviewer nHJa for more comprehensive results on computational complexity and efficiency.
>
> (v) Limitation and potential negative societal impact.
>
> In fact, we analyze the broader impacts and limitations of our work in Sec. 4 and Sec. 5 of the supplementary materials. Regrettably, limited by the constraints imposed on paper length, we apologize for placing these pivotal sections in the supplementary materials.

---

### Official Review · Reviewer_nHJa · 2023-07-08

**Soundness:** 3 good
**Presentation:** 3 good
**Contribution:** 3 good
**Rating:** 6
**Confidence:** 4

**Summary:**

This paper proposed a pyramid reconstruction network for tone mapping via a combination of existing lookup table and local Laplacian filter techniques, by utilizing closed-form Laplacian pyramid decomposition and reconstruction.

Specifically, the image-adaptive 3D LUTs is employed to manipulate the tone in the low-frequency image by leveraging the specific characteristics of the frequency information. The local Laplacian filter is utilized to refine the edge details in the high-frequency components in an adaptive manner.

Extensive experimental results on two benchmark datasets demonstrate that the proposed method performs favorably against state-of-the-art methods.


**Strengths:**

- Originality: The idea of optimize existing Laplacian pyramid decomposition and reconstruction based tone mapping method via the combination of lookup table and local Laplacian filter techniques is interesting and reasonable. Besides, the proposed image-adaptive learnable local Laplacian filter can preserve the local edge details.
- Clarity: This paper is well-organized and easy to follow.


**Weaknesses:**

- The algorithm description is not clear enough：\textit{e.g.,} the definition of objective function for the image-adaptive learnable local Laplacian filter in Eq. (4) is missing.

- Insufficient experimental analysis：
  - The runtime analyse are not provided.
  - Quantitative comparison of different methods on HDR+ datasets is missing.


**Questions:**

- How about the performance for the setting with image-adaptive Local Laplacian Filter disabled? This result helps to determine the effectiveness of the proposed technology.
- The visual comparison in Fig. 4 is not easy to understand. What does the heat map in the upper left corner mean?

---

> ### Author Rebuttal · Authors · 2023-08-06
>
> Thanks for your valuable comments. Now we respond to your concerns one by one.
>
> (i) Unclear algorithm description.
>
> The formula of the objective function for the image-adaptive learnable local Laplacian filter is similar to Eq. (3), albeit with the distinction that the parameters within it are learnable by a compact network. We will further refine this aspect in our final version to enhance its accessibility and clarity for readers.
>
> (ii) Runtime analyses.
>
> We draw on prior literature that only provides the number of parameters. To better characterize our model's computational complexity and efficiency, we provide a more diverse set of metrics, including the number of parameters, the multiply-accumulate operations (MACs), and the runtime. We will include these metrics in the final version of our paper and present them in subsequent tables:
>
> | Methods   | $\#$Params | MACs      | Runtime   |
> |-----------|------------|-----------|-----------|
> | UPE       | 999K       | 1.146G    | 8.42ms    |
> | HDRNet    | 482K       | 1.103G    | 4.56ms    |
> | CSRNet    | 37K        | 52.843G   | 48.28ms   |
> | DeepLPF   | 1.72M      | 454.422G  | 386.62ms  |
> | LUT       | 592K       | 0.676G    | 2.67ms    |
> | sLUT      | 4.52M      | 20.316G   | 9.63ms    |
> | CLUT      | 952K       | 9.391G    | 8.35ms    |
> | [3]       | 4.45M      | 1410.048G | 162.23ms  |
> | Ours      | 731K       | 2.923G    | 20.51ms   |
>
> All presented results are obtained using a 32GB NVIDIA V100 GPU. Runtime values are averaged over 1000 images, each with dimensions of $3840\times2160\times3$. Similarly, Multiply-Accumulate Operations (MACs) are computed based on the input dimensions $3840\times2160\times3$.
>
> The results show that the extensive computational demands of CSRNet, DeepLPF, and [3] (Reviewer zrx6 requires a comparison with [3]) lead to slow processing. These results can be attributed to the fact that they are pixel-level methods, rendering them more reliant on hardware capabilities, especially when processing large-resolution images. Conversely, our approach makes a trade-off between computational complexity and performance.
>
> (iii) Quantitative comparison of different methods on HDR+ datasets is missing.
>
> In fact, Table. 2 in our main manuscript provides a quantitative comparative analysis of the HDR+ dataset.
>
> (iv) Experiment about disable the image-adaptive Local Laplacian Filter.
>
> In fact, as described in Sec. 3.4, we have performed ablation experiments on each component of the model. Table. 3 and Fig. 6 provide the evaluation results of disabling the image-adaptive local Laplace filter qualitatively and quantitatively, respectively.
>
> (v) Why heat map in the upper left.
>
> It is imperative to clarify that the visual representation in Fig. 4 and Fig. 5 is an error map rather than a heat map. This error map represents the relationship difference derived by subtracting the reference image from the tone-mapped image. The rationale for this presentation lies in the central goal of the tone mapping task - to primarily recalibrate the tone of the image while compressing the dynamic range. Due to these common goals, the visual differences between the results produced by the various state-of-the-art methods are minimal. Therefore, the purpose of utilizing this error map is to facilitate a more precise identification of performance differences. Our final version will add some clarification to avoid any potential misunderstandings.
>
> Reference:
>
> [3] Vinker, Yael, Inbar Huberman-Spiegelglas, and Raanan Fattal. "Unpaired learning for high dynamic range image tone mapping." Proceedings of the IEEE/CVF International Conference on Computer Vision. 2021.

---

> > ### Comment · Reviewer_nHJa · 2023-08-19
> >
> > Thank you for the rebuttal. All my concerns were well addressed.

---

### Author Rebuttal · Authors · 2023-08-08

We provide qualitative comparison results with [1] and [3] (Reviewer zrx6 requires a comparison with [1]-[3]) in Fig. 1-4 in the attached PDF. The comparison results of [2] are not included since its source code is not publicly available. Despite efforts to contact the authors for assistance, no response was received within the rebuttal period. Thus, reproducing their work was unfeasible at this stage. However, our commitment to reproducing [2] remains, and we intend to include comparative results in our final version.

To better demonstrate the superiority of our methods, we also provide the quantitative comparison results of all methods on TMQI, Fidelity, and Naturalness metrics in the following table:

| Methods | PSNR  | SSIM  | LPIPS | Δ E   | TMQI  | Fidelity | Naturalness |
|---------|-------|-------|-------|-------|-------|----------|-------------|
| UPE     | 23.33 | 0.852 | 0.150 | 7.68  | 0.8789| 0.8951   | 0.4213      |
| HDRNet  | 24.15 | 0.845 | 0.110 | 7.15  | 0.8768| 0.8968   | 0.4212      |
| CSRNet  | 23.72 | 0.864 | 0.104 | 6.67  | 0.8880| 0.8908   | 0.4529      |
| DeepLPF | 25.73 | 0.902 | 0.073 | 6.05  | 0.8752| 0.8820    | 0.4419      |
| LUT     | 23.29 | 0.855 | 0.117 | 7.16  | 0.8818| 0.9018   | 0.4119      |
| sLUT    | 26.13 | 0.901 | 0.069 | 5.34  | 0.8854| 0.9017   | 0.4486      |
| CLUT    | 26.05 | 0.892 | 0.088 | 5.57  | 0.8863| 0.9020    | 0.4705      |
| [1]     | 18.52 | 0.799 | 0.168 | 12.20 | 0.8862| 0.8975   | 0.5092      |
| [2]     | N.A.  | N.A.  | N.A.  | N.A.  | N.A.  | N.A.     | N.A.        |
| [3]     | 16.21 | 0.709 | 0.215 | 17.60 | 0.8893| 0.9061   | 0.4723      |
| Ours    | 26.62 | 0.907 | 0.063 | 5.31  | 0.8977| 0.9089   | 0.5134      |

N.A. represents no public source code for [2] is accessible. All these results were evaluated on the HDR+ 480p dataset. The table above shows that TMQI produces similar results for [1], [3], and our approach. However, as shown in Fig. 1-4, visual results show substantial differences, emphasizing the need for a comprehensive assessment combining all metrics to express performance differences.

[1] Liang, Zhetong, et al. "A hybrid l1-l0 layer decomposition model for tone mapping." Proceedings of the IEEE conference on computer vision and pattern recognition. 2018.

[2] Su, Chien-Chuan, et al. "Explorable tone mapping operators." 2020 25th International Conference on Pattern Recognition (ICPR). IEEE, 2021.

[3] Vinker, Yael, Inbar Huberman-Spiegelglas, and Raanan Fattal. "Unpaired learning for high dynamic range image tone mapping." Proceedings of the IEEE/CVF International Conference on Computer Vision. 2021.

---

### Decision · Program_Chairs · 2023-09-21

**Decision:**

Accept (poster)

**Comment:**

Using closed-form Laplacian pyramid decomposition and reconstruction, this paper proposes a pyramid reconstruction network for tone mapping by combining existing lookup tables and local Laplacian filter techniques. In the initial reviews, reviewers raised concerns and offered suggestions regarding deeper analysis, the integration of the transformer model and fusion block into all pyramid levels, the number of parameters, evaluation on more benchmarks, and more evaluation metrics. In the rebuttal, further comparisons are made with SOTA methods, more metrics are provided for evaluation, runtime analysis is provided, and detailed descriptions are presented. Additionally, it explains why the PPR10K dataset was not adopted, and how the reported metrics differ from those in the original papers. A majority of the reviews are positive about the paper after the discussion period.